# Neonatal brain dynamic functional connectivity in term and preterm infants and its association with early childhood neurodevelopment

Lucas G. S. França [1,2,3], Judit Ciarrusta[1,2], Oliver Gale-Grant[1,2], Sunniva Fenn-Moltu[1,2], Sean Fitzgibbon[4], Andrew Chew [2], Shona Falconer[2], Ralica Dimitrova[1,2], Lucilio Cordero-Grande[2,5,6], Anthony N. Price[2], Emer Hughes[2], Jonathan O'Muircheartaigh [1,2,7], Eugene Duff[4,8,9], Jetro J. Tuulari[10,11,12], Gustavo Deco [13,14,15,16], Serena J. Counsell[2], Joseph V. Hajnal [2], Chiara Nosarti [2,17], Tomoki Arichi [2,7,18,19], A. David Edwards [2,7,20], Grainne McAlonan [1,20] & Dafnis Batalle [1,2,20] ✉

Brain dynamic functional connectivity characterises transient connections between brain regions. Features of brain dynamics have been linked to emotion and cognition in adult individuals, and atypical patterns have been associated with neurodevelopmental conditions such as autism. Although reliable functional brain networks have been consistently identified in neonates, little is known about the early development of dynamic functional connectivity. In this study we characterise dynamic functional connectivity with functional magnetic resonance imaging (fMRI) in the first few weeks of postnatal life in term-born ($n = 324$) and preterm-born ($n = 66$) individuals. We show that a dynamic landscape of brain connectivity is already established by the time of birth in the human brain, characterised by six transient states of neonatal functional connectivity with changing dynamics through the neonatal period. The pattern of dynamic connectivity is atypical in preterm-born infants, and associated with atypical social, sensory, and repetitive behaviours measured by the Quantitative Checklist for Autism in Toddlers (Q-CHAT) scores at 18 months of age.

Resting-state functional magnetic resonance imaging (fMRI) is often used to infer how different areas of the brain function together[1] and to establish whole-brain functional connectivity networks[2], assuming a condition of stationarity. While this is a useful approach to ascertain "on average" characteristics of brain activity, connectivity of the brain is intrinsically dynamic, i.e., non-stationary[3,4]. Addressing this problem, dynamic functional connectivity (dFC) measures the constant neural adjustments needed to control different brain states, adapt to transient situations, and integrate information[5]. The dynamical properties characterising the continuous shifting between connectivity profiles or "states" have been linked to processes such as language[6,7], cognition[8–11], and motor function[12,13]. Importantly, altered brain dynamics have also been linked to the clinical features and/or cognitive dysfunction in neurodevelopmental conditions such as schizophrenia[3], attention deficit hyperactivity disorder[14] and autism spectrum disorder (ASD)[15]. Individuals with ASD, for example, have

been reported to switch between different connectivity profiles more directly, whereas typically developing individuals switch between those same brain states via an intermediate connectivity profile[15]. However, although it is increasingly appreciated that neurodevelopmental conditions likely have their origins in the perinatal period, little is known about the brain's dynamic properties at this critical juncture. Moreover, it is also not clear whether neonatal dFC characteristics are associated with later childhood neurodevelopment, and in particular which of these characteristics signal a higher likelihood of later neurodevelopmental difficulties.

Conventional studies of "static" functional brain connectivity in early life have shown that the spatial representation of resting state networks (RSNs) appears to be relatively mature and adult-like even soon after birth[16–18], with mature primary RSNs and emerging association RSNs, consistent with a primary-to-higher-order ontogenetic sequence of brain development[19]. It has also been confirmed that these RSNs may be disrupted by perinatal exposures, such as preterm birth[16,19–21], which increase the likelihood of neurodevelopmental impairments[22]. However, despite the evidence for the importance of dynamics in older groups, the dynamics of these networks in early life remain to be described.

In this study, we applied state-of-the-art techniques to evaluate fMRI in a cohort of term-born ($n = 324$) and preterm-born babies ($n = 66$) scanned at term equivalent age from of the developing Human Connectome Project (dHCP), the largest publicly available population-based dataset of the healthy new-born brain[23]. We used two methods that tap into dynamic brain function. First, we characterised global dynamics using Kuramoto Order Parameter (KOP)[24] based measures, namely mean synchronisation and metastability[25,26] as global measures of brain synchronicity and flexibility[25,27]. Second, we characterised modular dynamics. That is, we identified sub-networks involved in temporal "states", i.e., paroxysmal modes representing synchronisation of the brain, using Leading Eigenvector Analysis (LEiDA) which is a time-resolved metric[28,29]. We assessed whether neonatal brain state features (fractional occupancy, dwell times, state mean synchronisation, and state metastability) and state transition probabilities were associated with postmenstrual age (PMA) at scan, postnatal days (PND) at scan and preterm-birth; and whether they correlate with neurodevelopmental outcomes at 18 months measured using the Bayley Scales of Infant and Toddler Development, 3rd Edition (Bayley-III)[30], and atypical social, sensory and repetitive behaviours measured by the Quantitative Checklist for Autism in Toddlers (Q-CHAT)[31]. This allowed us to test three hypotheses: 1) that neonatal brain dynamics quickly develop with age at scan; 2) that preterm birth alters the typical pattern of functional brain dynamics; and 3) that neonatal brain dynamics are linked to neurodevelopmental and behavioural outcome measures at age 18 months.

## Results

### Global brain dynamics

We first assessed the association of global dynamic features with age and postnatal days at scan (PMA and PND at scan, respectively) in term-born individuals only ($n = 324$). We did not observe any associations between PMA at scan and global dynamic features. Nevertheless, there was an association between PND at scan with metastability ($t = -2.4$; $p = 0.017$, Table 1). In comparison to term-born infants, preterm-born infants had lower mean synchronisation (Cohen's $D = 0.567$−medium effect size; $p < 0.001$) and metastability ($D = 0.454$−medium effect size; $p < 0.001$) (Fig. 1a, b, Table 1). We then analysed associations between global dynamic features (mean synchronisation and metastability) and neurodevelopmental markers, namely Q-CHAT and Bayley-III scores for the entire cohort (term- and preterm-born, $n = 390$). Across the whole cohort, there was no significant association between metastability and any neurodevelopmental outcome, although mean synchronisation showed a weak but significant association with Q-CHAT scores ($t = -2.6$; $p = 0.011$), i.e., lower neonatal mean synchronisation was associated with higher Q-CHAT scores, indicative of more atypical social, sensory, and repetitive behaviours at 18 months of age. Results were robust to the choice of atlas parcellation, as similar findings were obtained with the Melbourne Children's Regional Infant Brain (M-CRIB) atlas (Supplementary Table S1), including a significant association of metastability with PND at scan ($p = 0.019$), and reduced mean synchronisation and metastability in preterm-born infants when compared with term counterparts (Cohen's $D = 0.628$, $p < 0.001$; and Cohen's $D = 0.480$, $p < 0.001$ respectively, see Supplementary Table S1). Analogously, M-CRIB's mean synchronisation was also significantly associated with Q-CHAT scores ($t = -2.7$, $p = 0.009$).

### Neonatal brain states

We defined six different brain states, obtained heuristically from K-Means clustering, using the LEiDA approach[28,29] (Fig. 2). Briefly, we averaged the fMRI timeseries into 90 cortical and subcortical regions defined by the Anatomical Automated Labels (AAL) atlas adapted to the neonatal brain[32]. We estimated the phase synchronisation between each pair of parcels, obtaining a dynamic functional connectivity matrix for each fMRI timepoint[33]. We calculated the first eigenvector of each matrix, for each instant of time, and clustered those with K-Means with $K = 6$ (see more details in Methods section). Three of the six states we identified showed widespread phase concordance amongst brain parcels, we refer to these as whole-brain global synchronisation states, namely global state A, global state B, and global state C. Three states were more regionally constrained: one state showed synchronous phases in the occipital cortex (Occ. State); one state represented high synchrony for regions mainly in the sensorimotor cortex (SM State); and one state comprised high levels of synchronisation in the frontal cortex, angular gyrus, and posterior cingulate gyrus (for simplicity we refer to this as frontoparietal or FP state).

### Landscape of modular brain dynamics in term-born neonates

We compared the main dynamic features of the six identified states in term-born participants ($n = 324$) via mean synchronisation, metastability, fractional occupancy and dwell times markers for each state. There were significant differences between states in mean

---

**Table 1 | Association of global dynamic features (synchronisation and metastability) with PMA and PND at scan, and association with preterm birth**

| | Term ($n = 324$) | | | | Term vs Preterm ($n = 390$) | | | |
| | PMA at scan | | PND at scan | | Term ($n = 324$) | Preterm ($n = 66$) | Cohen's $D$ | $p$ value[b] |
| | $t$[a] | $p$ value[a] | $t$[a] | $p$ value[a] | [mean (S.D.)] | | | |
|---|---|---|---|---|---|---|---|---|
| Mean synchronisation | 0.039 | 0.970 | 1.143 | 0.259 | 0.52 (0.08) | 0.48 (0.08) | 0.567 | $p < 0.001$[c] |
| Metastability | 0.877 | 0.379 | −2.403 | 0.017[c] | 0.20 (0.02) | 0.19 (0.02) | 0.454 | $p < 0.001$[c] |

$p$ values obtained with a two-sided permutation test.
[a]GLM1 (including 324 term-born babies): $y \sim \beta_O + \beta_1 PMA + \beta_2 PND + \beta_3 Sex + \beta_4 Motion$ outliers (FD).
[b]GLM2 (including 324 term-born and 66 preterm-born babies): $y \sim \beta_O + \beta_1 Preterm\text{-}born + \beta_2 PMA + \beta_3 Sex + \beta_4 Motion$ outliers (FD).
[c]Indicates results surviving FDR multiple comparison correction with α error at 5%.

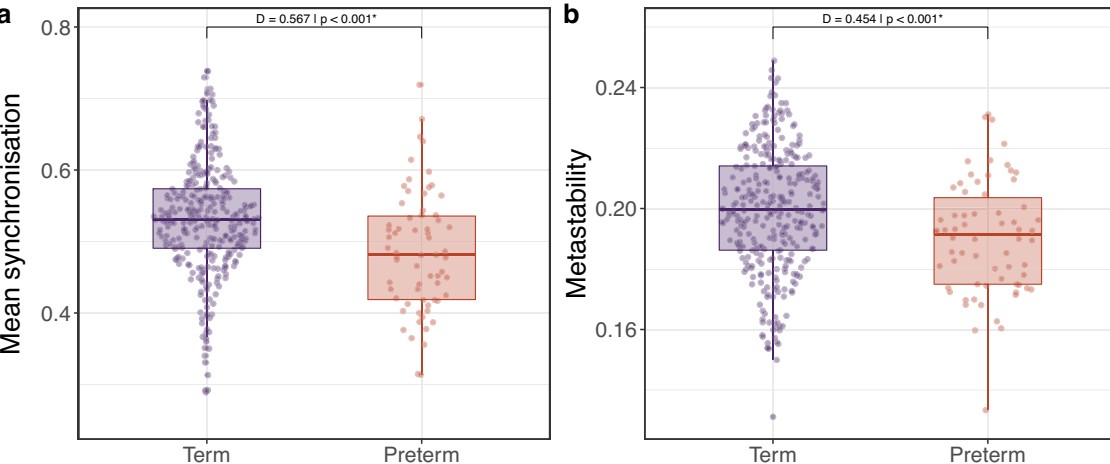

**Fig. 1** | Association of preterm birth with (**a**) mean synchronisation and (**b**) metastability. *$p$ values obtained with a two-sided permutation test—results surviving FDR multiple comparison correction with α error at 5%. $D$: Cohen's effect size. Term ($n = 324$) and preterm ($n = 66$) born individuals. Boxplots showing 0th, 25th, 50th, 75th, and 100th centiles. Outliers defined when value larger than 1.5*IQR + 75th centile. $D$: Cohen's D. Source data are provided as Source Data file.

synchronisation, metastability, fractional occupancy, and dwell times when tested using a Type III ANOVA model with Satterthwaite's method: $F(5, 1615) = 13163$; $p < 0.001$ for mean synchronisation, $F(5, 1615) = 291$; $p < 0.001$ for mean metastability, $F(5, 1938) = 514$; $p < 0.001$ for mean fractional occupancy, and $F(5, 1938) = 734$; $p < 0.001$ for mean dwell times. Between state effects showed that global state A had increased mean synchronisation, mean fractional occupancy, and mean dwell times when compared with the other five states. Moreover, global states B and C also had a heightened mean synchronisation which was intermediate between the values recorded for global state A and the other states (see Supplementary Fig. S1 for between-group effects statistics). We characterised the normative brain state transition probabilities landscape in Fig. 3a. Most occurrences are those of dwelling sequences, i.e., repeated continuous occurrences of the same state, with probabilities above 89% for all six states. Excluding those dwelling sequences, a complex dynamic profile is displayed by the 12 most frequent transitions. For instance, the brain transitions to and from global state A (which has high levels of synchronisation), via global states B and C (which have intermediate levels of synchronisation), while brain transitions into FP state occur through both global and regionally constrained states: global state B, occipital state, and SM state.

### Modular brain state features associated with age at scan in term-born neonates

Higher PMA at scan was positively correlated with increased dwell times ($t = 4.4$; $p < 0.001$), increased fractional occupancy ($t = 5.3$, $p < 0.001$), and increased mean synchronisation ($t = 3.6$; $p < 0.001$) for global state C; and with increased fractional occupancy ($t = 2.8$, $p < 0.001$) and mean synchronisation in the sensorimotor state ($t = 3.1$; $p = 0.002$). PMA at scan was negatively correlated with increased fractional occupancy in the global state B ($t = -2.8$; $p = 0.004$). Postnatal age (PND at scan) was associated with shorter dwell times in the global state B ($t = -2.8$; $p = 0.004$), occipital state ($t = -2.5$; $p = 0.013$), and sensorimotor state ($t = -2.8$; $p = 0.006$). Significant associations of brain state features with PMA and PND at scan that survive False Discovery Rate (FDR) multiple comparison are summarised in Fig. 3c, d; and compared in Fig. 4g.

### Brain state transitions associated with age at scan in term-born neonates

Higher PMA at scan was positively associated with an increased likelihood of transitioning from global state A to C ($t = 3.2$; $p = 0.002$),

from global state B to frontoparietal state ($t = 3.1$; $p = 0.002$), and to stay (dwell) in global state C ($t = 3.2$; $p = 0.001$). Older PMA was also associated with a lower likelihood of transitioning from global state C to occipital state ($t = -2.8$; $p = 0.006$); and lower probability of staying in global state B ($t = -2.9$; $p = 0.005$). Significant associations of state transition probabilities with PMA at scan are shown in Fig. 3e. Increased postnatal age (PND at scan) was associated with increased transitions from frontoparietal and occipital states into global state B [($t = 3.1$; $p = 0.002$) and ($t = 3.3$; $p = 0.001$), respectively]; increased transition from the occipital state into state C ($t = 3.1$; $p = 0.002$); a reduction of transition likelihood from state B into a frontoparietal state ($t = -3.0$; $p = 0.003$); and a reduction of probability to stay (dwell) in state C ($t = -2.7$; $p = 0.007$), occipital state ($t = -3.4$; $p = 0.001$), and sensorimotor state ($t = -2.7$; $p = 0.007$). Significant associations of state transition probabilities with PND at scan, all surviving FDR multiple comparison corrections, are summarised in Fig. 3f. For a comparison of the distinct association with age (PMA at scan) and postnatal experience (PND at scan) in brain state transitions see Fig. 4h. Age at scan and postnatal experience had distinct correlates: for example, transitions from intermediate whole-brain synchronisation state C to the occipital state decreased with PMA at scan, while transitions in the opposite direction (from occipital state to synchronisation state C) were increased with PND at scan.

### Atypical modular brain state features in preterm born neonates

Compared to term-born participants, preterm-born babies had shorter dwell times for the global state A ($t = -4.6$; $p < 0.001$, Fig. 4a); decreased fractional occupancy for the global state A ($t = -4.1$; $p < 0.001$); and increased fractional occupancy for global state B ($t = 3.4$; $p = 0.001$), occipital state ($t = 2.2$; $p = 0.030$), and frontoparietal state ($t = 2.7$; $p = 0.009$) (Fig. 4b). Preterm birth was also associated with lower mean synchronisation of global state A ($t = -5.1$; $p < 0.001$), global state B ($t = -3.4$; $p = 0.001$), global state C ($t = -2.8$; $p = 0.005$), occipital state ($t = -2.2$; $p = 0.031$), and frontoparietal state ($t = -2.6$; $p = 0.010$)) (Fig. 4c); and reduced metastability for the global state A ($t = -2.5$; $p = 0.014$) and frontoparietal state ($t = -4.3$; $p < 0.001$) (Fig. 4d).

### Association with preterm birth on brain dynamics and brain state transition probability

Preterm birth was associated with an increased transition towards an occipital connectivity profile, i.e., an increased transition probability from global state A to C ($t = 4.7$; $p < 0.001$) and from global state C to

occipital ($t = 3.1$; $p = 0.002$); as well as a reduction in the probability staying (dwelling) in global state A ($t = -4.4$; $p < 0.001$); see Fig. 4f. A similar profile of results was obtained when assessing how brain dynamic features changed with gestational age at birth which essentially captures preterm and term birth in a continuous way, see Supplementary Figure S2. Significant changes in transition probabilities associated with preterm-birth and their relation to associations with PMA and PND at scan are summarised in Fig. 4h.

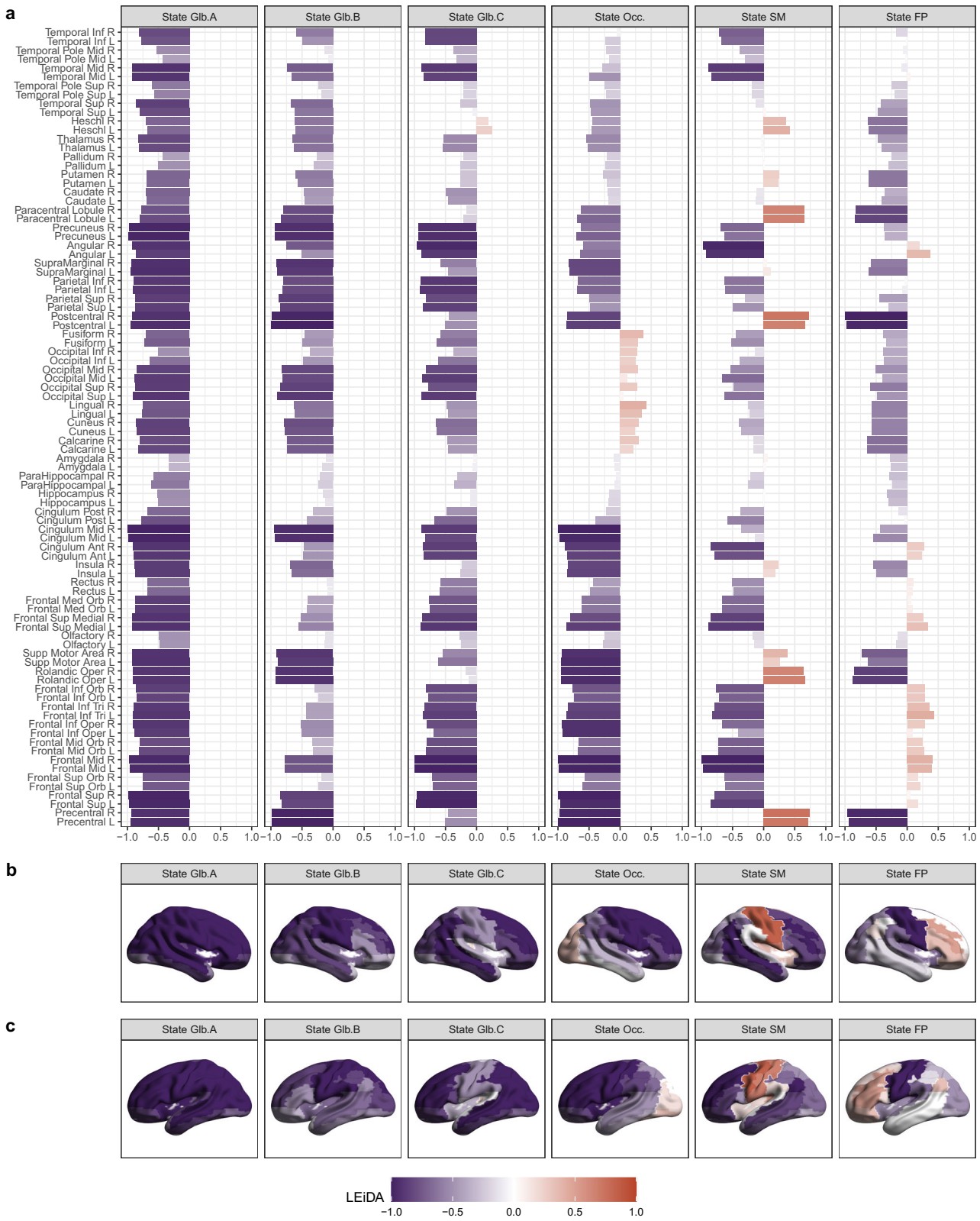

**Fig. 2 | Transient brain states in neonates.** Ordered from left to right by level of global synchronicity. **a** LEiDA vectors for each of the six brain states identified in the neonatal brain using AAL parcels. **b** Representation of LEiDA on brain surfaces (right side view). **c** Representation of LEiDA on brain surfaces (left side view). Glb. Global, Occ. Occipital, SM Sensorimotor, FP Frontoparietal. Source data are provided as Source Data file.

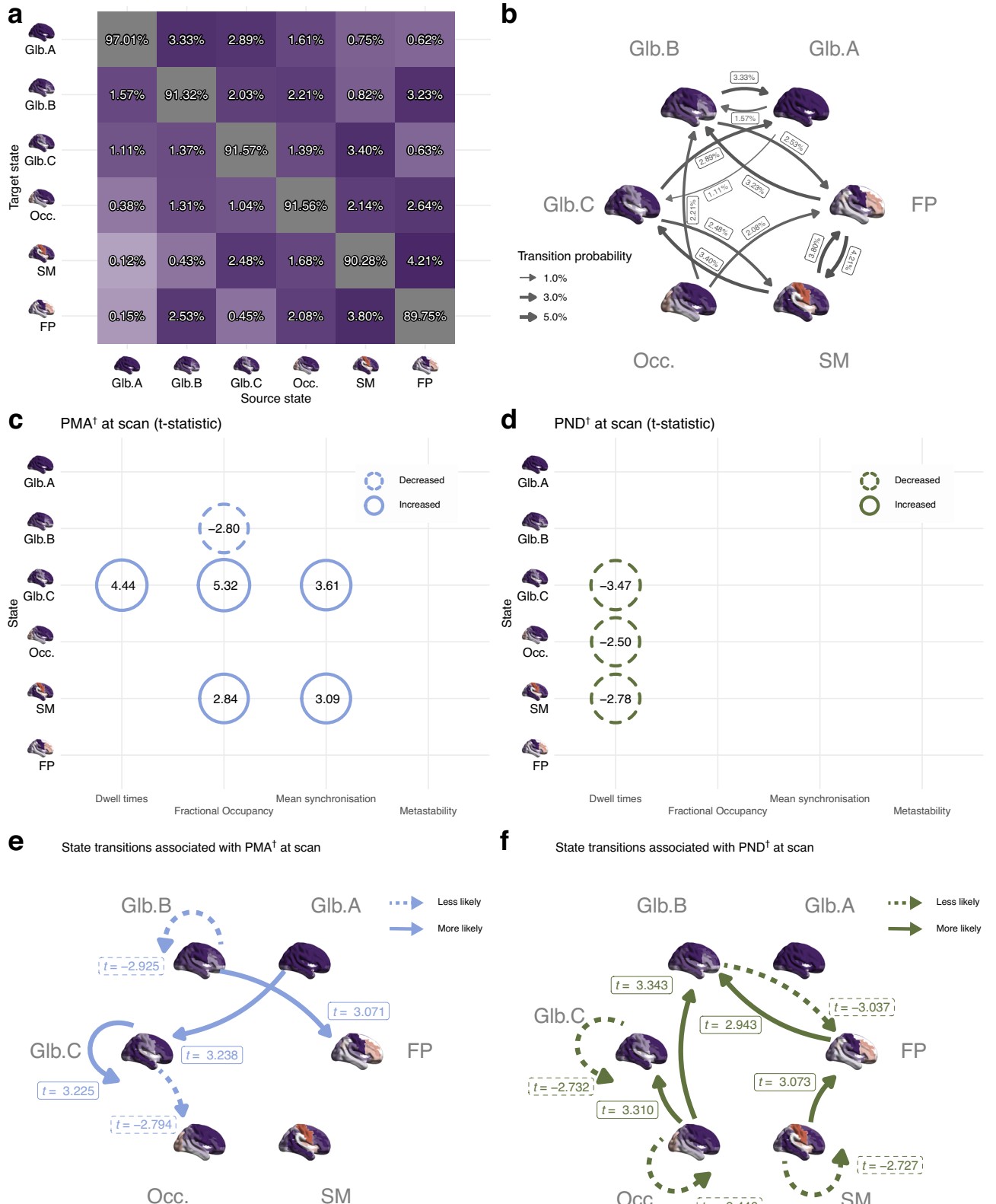

**Fig. 3 | Brain dynamics in term-born neonates ($n$ = 324). a** All transitions including dwelling state sequences. **b** Main transitions (top 12) between states excluding dwelling sequences. **c** Summary of brain state features significantly associated with PMA at scan. **d** Summary of brain state features significantly associated with PND at scan. **e** Summary of significant correlations between state transitions probabilities and PMA at scan. **f** Summary of significant correlations between state transitions probabilities and PND at scan. †GLM1 (including 324 term-born babies): $y - \beta_0 + \beta_1 PMA + \beta_2 PND + \beta_3 Sex + \beta_4 Motion$ outliers (FD). Values shown in (**c**) and (**d**) indicate t-statistics. All significant associations (two-sided permutation test) shown in this figure survive FDR multiple comparison correction with α error at 5%. Glb. Global, Occ. Occipital, SM Sensorimotor, FP Frontoparietal, PMA Postmenstrual age, PND Postnatal days. Source data are provided as Source Data file.

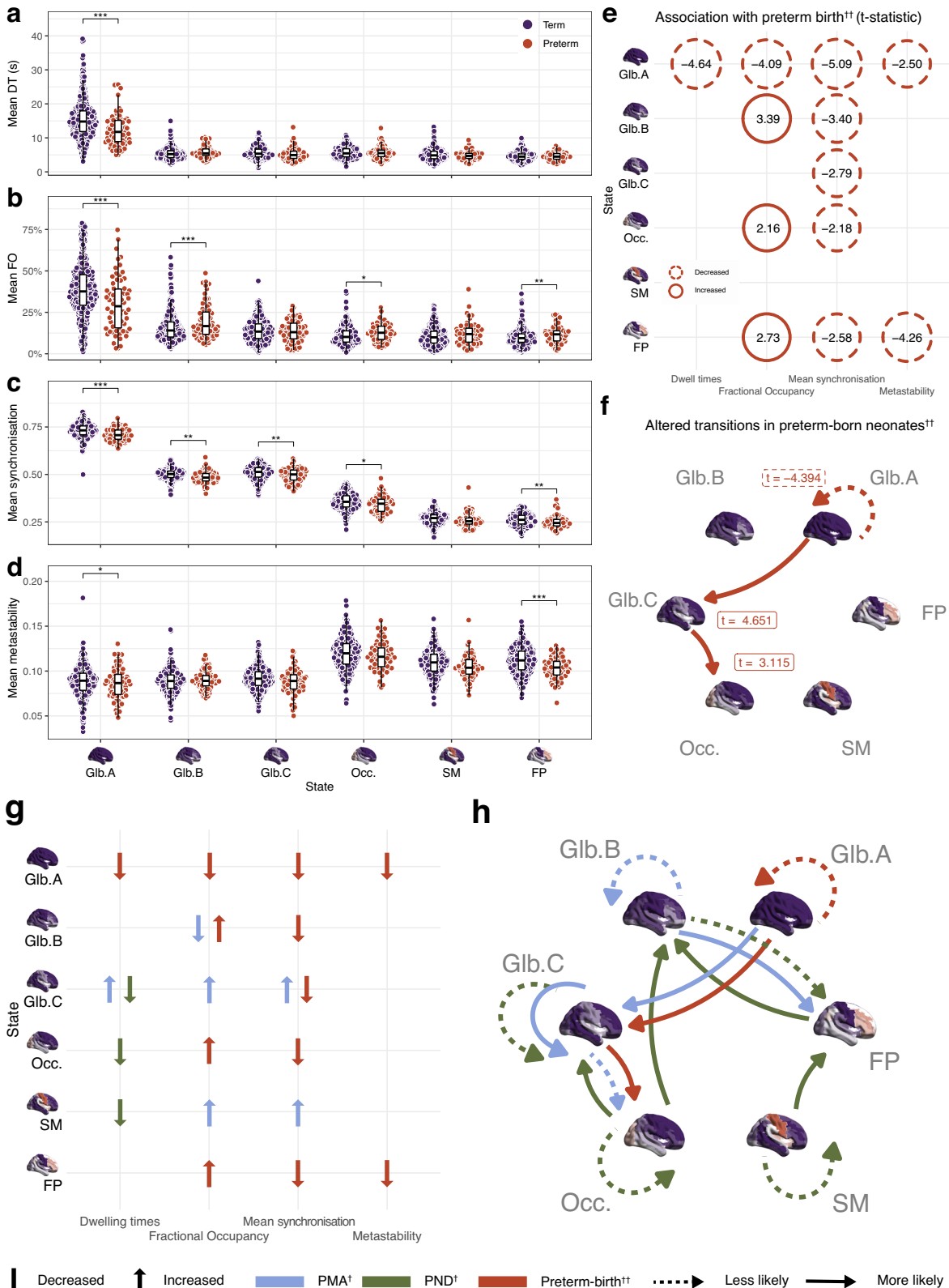

**Fig. 4 | Brain dynamics in preterm-born neonates (n = 390). a** Mean dwell times (DT). **b** Mean fractional occupancy (FO). **c** Mean synchronisation. **d** Metastability. **e** Summary of significant associations with preterm birth. **f** Association of state transitions probabilities and preterm birth. **g** Summary of brain state features significantly associated with preterm birth, and comparison with those significantly associated with PMA and PND at scan. **h** Summary of brain state transition probabilities associated with increased PMA, increased PND, and/or preterm birth. [†]GLM1 (324 term-born babies): $y - \beta_O + \beta_1 PMA + \beta_2 PND + \beta_3 Sex + \beta_4 Motion\ outliers\ (FD)$. [††]GLM2 and (**a**) (**b**) (**c**) (**d**) (324 term-born and 66 preterm-born babies): $y - \beta_O +$

$\beta_1 Preterm\text{-}born + \beta_2 PMA + \beta_3 Sex + \beta_4 Motion\ outliers\ (FD)$. * $p < 0.05$. ** $p < 0.01$. *** $p < 0.001$ obtained with a two-sided permutation test. Values shown in (**e**) indicate t-statistics. Boxplots showing 0th, 25th, 50th, 75th and 100th centiles. Outliers defined when value larger than 1.5*IQR + 75th centile. All significant associations highlighted survive FDR multiple comparison correction with α error at 5%. Glb. Global, Occ. Occipital, SM Sensorimotor, FP Frontoparietal, PMA Postmenstrual age, PND Postnatal days, DT Dwell time, FO Fractional occupancy. Source data are provided as Source Data file.

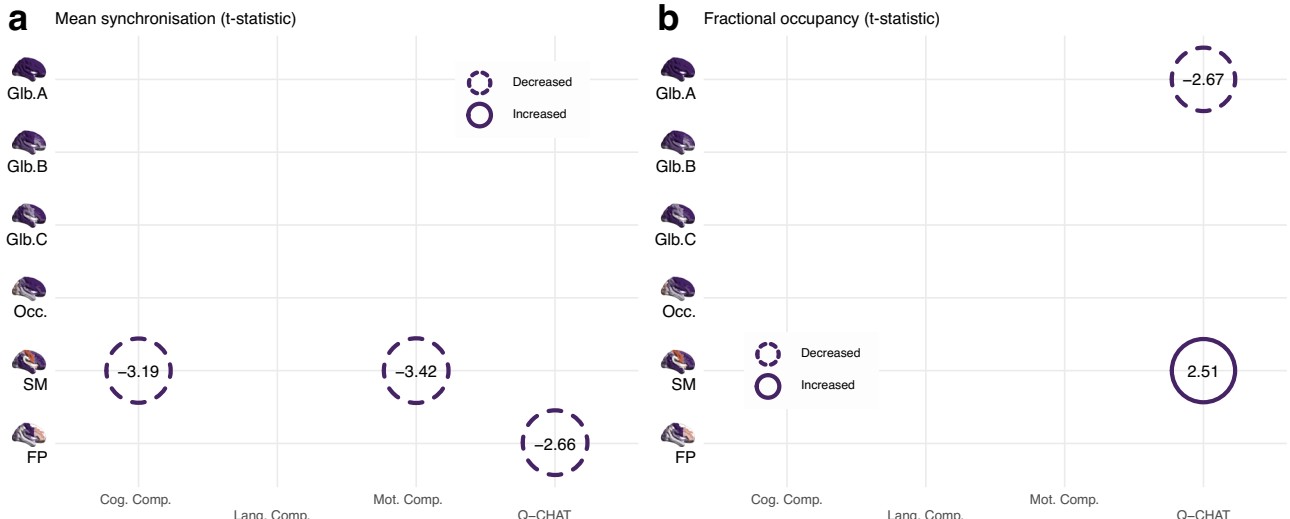

**Fig. 5 | Summary of associations of brain state features with neurodevelopmental outcomes at 18 months corrected age.** Association of average (**a**) mean synchronisation and (**b**) fractional occupancy in each of the six defined brain states during perinatal period with cognitive, language and motor Bayley-III composite scores and Q-CHAT scores. GLM3 (257 term-born and 48 preterm-born babies for Bayley-III; and 254 term-born and 46 preterm-born babies for Q-CHAT): y $-\beta_0 + \beta_1$GA + $\beta_2$PMA + $\beta_3$Sex + $\beta_4$Motion outliers (FD) + $\beta_5$[Corrected age at assessment] + $\beta_6$[Assessed component] + $\beta_7$[Index of multiple deprivation]. Values shown in both panels indicate t-statistics. All significant associations (two-sided permutation test) highlighted survive FDR multiple comparison correction with α error at 5%. Glb. Global, Occ. Occipital, SM Sensorimotor, FP Frontoparietal, Cog. Cognitive, Lang. Language, Mot. Motor, Comp. Component, Q-CHAT Quantitative Checklist for Autism in Toddlers.

## Neonatal brain states and early childhood neurodevelopment

Lastly, we investigated associations between brain states dynamics after birth and neurodevelopmental outcomes at 18 months of age measured with Bayley-III and Q-CHAT scores. Higher mean synchronisation of the sensorimotor state in neonates was associated with poorer performance on both Bayley's cognitive and motor scores assessed at 18 months of age ($t = -3.2$, $p = 0.002$; and $t = -3.4$, $p = 0.002$ respectively; Fig. 5a). Higher Q-CHAT scores at 18 months of age were associated with reduced mean synchronisation of frontoparietal state ($t = -2.7$, $p = 0.008$), higher fractional occupancy of sensorimotor state ($t = 2.5$, $p = 0.014$), and reduced fractional occupancy of global state A ($t = -2.7$, $p = 0.010$; Fig. 5b).

## A similar pattern of regional brain dynamics was obtained with an alternative parcellation scheme

The six brain states obtained with the alternative M-CRIB neonatal atlas were compatible with the ones obtained for the AAL atlas (Supplementary Fig. S3), with three global synchronisation states and three more regionally constrained. States 1, 2, and 3 obtained with the M-CRIB atlas were consistent with global synchronisation states obtained with the AAL atlas. The M-CRIB's State 4 featured some of the structures present in occipital state, State 5 featured similar structures to the sensorimotor state, and State 6 was concordant with FP state obtained from AAL.

Moreover, for the analyses using the M-CRIB atlas in term-born babies (Supplementary Fig. S4), PMA was also associated with increased fractional occupancy ($t = 2.9$; $p = 0.002$) and mean synchronisation ($t = 3.0$; $p = 0.003$) for State 5 (-SM); and PND was associated with reduced dwell times ($t = -2.5$; $p = 0.012$) for State 5 (-SM). Transitions from States 6 (-FP) to State 2 (-Glb.) also increased with PND ($t = 3.3$; $p = 0.002$). Pre-term birth was also associated with similar changes in state metrics for M-CRIB atlas (Supplementary Fig. S5) with increased fractional occupancy for State 4 (-Occ.) ($t = 2.3$; $p = 0.020$) and State 6 (-FP) ($t = 3.0$; $p = 0.002$); and reduced mean synchronisation for State 6 (-FP) ($t = -2.2$; $p = 0.027$). See Supplementary Figure S6 for a similar analysis with GA.

Association with developmental outcomes in cognitive ($t = -4.0$; $p < 0.001$) and motor ($t = -3.4$; $p < 0.001$) components of Bayley-III showed negative associations with mean synchronisation for State 5 (-SM) and higher Q-CHAT scores were associated with reduced fractional occupancy for State 1 ($t = -2.6$; $p = 0.008$), see Supplementary Fig. S7 for a summary. A detailed description of results using M-CRIB atlas for parcellation is available in Supplementary Materials.

## Discussion

We used global and modular fMRI signal analysis tools to investigate the characteristics of dynamic functional connectivity in a large sample of term and preterm born neonates. This study characterises the fundamental features of the neonatal repertoire of brain states and their dynamics during early human development. We found that brain dynamics are disrupted in preterm-born infants and that the profile of brain dynamics in early postnatal life is associated with a range of early childhood neurodevelopmental and behavioural outcomes at 18 months of age.

We found that global dynamic features remained relatively stable in early postnatal development (37-44 weeks PMA at scan) in a term-born population, although lower metastability was observed with more postnatal days at scan, suggesting that ex-utero life experience reduces brain dynamic flexibility after birth, and promotes more stable connectivity patterns. This is consistent with our observation that preterm born infants scanned at term equivalent age, and thus with greater exposure to postnatal life experience at the time of scan, had lower metastability than term born children. However, preterm babies also had lower mean synchronisation at term-equivalent age suggesting a unique pattern of alteration in brain dynamics associated with preterm birth which is independent of the extent of ex-utero life exposure (PND at scan). Lower metastability has been previously associated with impairments in cognitive flexibility of mature individuals after traumatic brain injury[11]. However, the negative association between duration of postnatal life and metastability observed here is unlikely to relate to cognitive flexibility, but rather reflect a refinement of network dynamics driven by primary sensory stimulation associated

with ex-utero life experience. Metastability seems to be additionally reduced by term equivalent age in preterm born babies, which may be consistent with theories suggesting perinatal stress atypically accelerates brain maturation, with potentially negative long-term impact[34].

Our study extends prior work which has observed that, later in childhood, very preterm born infants have suboptimal neural synchrony and altered global dynamic connectivity patterns when scanned later in childhood[35]. Our metrics suggest that the association of preterm birth with brain dynamics may begin much earlier in life. Such differences in preterm-born babies may have their roots in alterations in the framework of structural and functional networks reported to accompany preterm birth. For example, previous studies have shown that preterm birth is associated with altered brain structure[36,37], global functional architecture[19,38], and structural network changes in the neonatal period[39] which continue to be present at school-age[40] and into young adulthood[41,42]. Global dynamic features were linked to later behaviour: lower mean synchronisation in the neonatal brain was associated with higher Q-CHAT scores at 18 months. Although high scores on the Q-CHAT indicate more autistic traits, in this study, the Q-CHAT captured a continuum of social, sensory, and repetitive behaviours across the normal distribution[31,43]. Thus, while global metrics might usefully signpost the trajectory of foundational dynamic steps of human brain development, their association with Q-CHAT should not be over-interpreted as our study was not a study of ASD and we did not examine children at an age where neurodevelopmental diagnoses begin to be formalised.

Summary metrics of global dynamics are likely themselves to be underpinned by much more complex activity. Therefore, we explored the emergence and behaviour of modular brain states in the early postnatal period. We characterised six transient states in the newborn brain at term equivalent age. Amongst the three states that showed widespread concordance (global states A–C), the first encompassed nearly all of the analysed cortical regions, the second showed a higher contribution of all sensory regions (auditory, sensorimotor, and visual) and the third state showed a higher contribution of visual and frontoparietal cortices. The other three states had a more restricted/regional span with distinct contributions from sensorimotor, visual and frontoparietal regions. The majority of these states thus encompassed primary sensory networks which are already known to mature earlier than higher order networks[16,19,44]. This adds confidence to the results reported in studies of dynamic FC in neonates around birth. For example, Ma et al. described dynamic functional connectivity with four brain states that encompassed default-mode, dorsal attention, auditory, sensorimotor, and visual networks in 37 term neonates[45]. Here we establish a series of six brain states and describe associations with age at scan and postnatal days in a larger cohort with 324 term neonates. We observed that in newborn infants, whole-brain synchrony state A had the highest mean synchronisation, as well as largest fractional occupancy and dwell times; thus suggesting that newborn infants spend a large amount of their time in a state of global phase synchrony which is a similar to the previously described dominant pattern of whole-brain synchronisation seen with both fMRI[45,46] and EEG[47]. Together with prior studies, our work supports the idea that large scale activity plays a crucial role in early brain development. Our results also align with the concept that this activity could support large scale cortical network formation and may foster the associated long-range connections, which are known to subsequently mature during the first postnatal year[48,49].

We observed a positive correlation of occupancy and mean synchronisation within the sensorimotor cortices with increasing PMA at scan. This supports existing evidence that this system is relatively mature, both in function and structure in comparison to other systems at birth. This may be the product of significant short-range functional reorganisation during the last foetal trimester[50,51] to support the functional specificity needed to respond to sensory information

coming from feet, hands, and mouth[52,53]. We also found that increased PMA at scan, was associated with an increased probability of transitioning into frontoparietal synchronicity states comprising the anterior part of the Default Mode Network[54]. This is concordant with other evidence that, although this system is relatively immature at birth, it undergoes significant changes postnatally with increasing recruitment of frontoparietal areas into the network[16,19]. Our work suggests that this system is recruited more and more with age in the postnatal period. There was also a significant association with PND at scan on an increasing probability of transition from occipital to whole-brain synchronisation states. This finding is in line with key stages of neurodevelopment. Specifically, changes in transitions from an occipital cortex profile may help the maturation of visuomotor abilities and sensory integration in early infancy in the environment outside the womb[55]. Finally, we observed increased transitions from sensorimotor to frontoparietal structures with increasing duration of postnatal life, perhaps reflecting the emerging of functional maturation of frontoparietal systems, which coincides with high interneuron migrations to these regions[56], relative to the already mature sensorimotor systems[51].

Preterm birth has been associated with a higher likelihood of atypical neurodevelopment[22] including a greater rate of autism diagnosis[57–62]. Previous studies from our group have shown preterm born infants have alterations in their functional architecture[16,19,38]. Here, we extend this work to report that dynamic functional connectivity is also altered in preterm born infants, showing increasing fractional occupancy of occipital and frontal states and increased transitions from global to occipital state; and decreased dwelling for whole-brain high synchronisation state. Only one other study has investigated the association of preterm birth with atypical dynamic functional connectivity[45]. They reported significantly shorter mean dwell times in a state with stronger connectivity between sensorimotor and auditory cortices and significantly higher mean dwell times for a global state[45]. Direct comparison with our findings is challenging however, given the higher resolution of our study in terms of a larger number of ROIs included but also the power available from our larger sample. We observed multiple global states in neonates with diverse contributions from occipital and frontoparietal regions and a negative association with prematurity on dwell times in state A. In summary, preterm born infants show shifts in dynamic functional connectivity towards occipital and frontoparietal synchrony profiles and suppresses whole-brain synchronisation modes. Preterm-birth is known to impact on cognition throughout the lifespan[63,64]. Our results raise the possibility, that alterations in brain dynamic functional connectivity that are present soon after birth have functional consequences.

Our work and others consistently recognise the neonatal period as a key time for sensorimotor cortical development[50,52,53] and subsequent transition to higher-order network connectivity[16,19]. We extended this observation here, to show that altered brain dynamics contribute to both general developmental outcomes and more specific social, sensory, and repetitive behaviours at age 18 months. Lower levels of synchrony in the sensorimotor state around term were positively correlated with better cognitive and motor outcomes (Bayley-III) at 18 months. This association is evident in the analyses with both the AAL and M-CRIB atlases. However, when there was lower fractional occupancy of the high whole-brain synchronisation state A and increased fractional occupancy of the sensorimotor state around birth, there were more atypical social, sensory, and repetitive traits present at 18 months, as captured by the Q-CHAT in AAL atlas. Thus, our work indicates that the link between brain dynamics and autistic traits is not only limited to state transitions in adulthood[15,65] but may comprise alterations in state occupancy and overall synchronicity as well as transitions established during early development. One possible explanation for the association between the sensorimotor cortex dynamics and later social, sensory and repetitive traits, is that altered brain dynamics in the neonatal period may predispose individuals to

exhibit unusual responses to sensory stimuli[66]. The positive correlation of higher fractional occupancy of the sensorimotor state and a higher Q-CHAT score could also represent an overreliance on that particular network during the neonatal period–which may impact upon the development of higher order networks.

We emphasise that we did not follow-up the children with higher Q-CHAT scores beyond 18 months, thus our work did not evaluate predictors of a confirmed diagnosis of ASD, nor is our study about diagnosed autism. However, children who go on to receive a diagnosis of autism and those with a broader phenotype may already show emerging traits of the condition. This was the basis for development of the Q-CHAT[31,43]. There is also accumulating evidence across different modalities that infants who go on to receive a diagnosis of autism have differences in their neurobiology and physiology from as young as 6–9 months[67–72] Thus our work links emerging social cognitive profile relevant to ASD, also to dynamic functional connectivity at birth, especially within sensory networks. This correlation between higher scores on an instrument, which captures early features relevant to ASD (though not necessarily diagnostic) to the dynamics of sensory systems, is in agreement with the importance of sensory processes throughout the lifespan in individuals who have an ASD diagnosis. Longitudinal studies are clearly needed, but our work adds to the evidence for a neurodiverse trajectory of sensory systems to autistic features across the lifespan. Sensory differences are among the first features to signal a diagnosis of ASD[70,73] and persist, remaining core to the diagnosis[74]. For example, we have previously reported that neonates with an increased familial likelihood of ASD had higher regional homogeneity in the sensorimotor cortex[75], which fits with the higher fractional occupancy we record in this region. Studies of diagnosed individuals have reported atypical activation of the motor cortex[76], which likely affects the translation of visual input into motor understanding, with a potential impact on social interaction. These functional brain differences in sensory systems are thought to arise from alterations in excitation-inhibition pathways, especially GABA neurotransmission. Evidence includes a tight relationship between sensory processing and differences in sensory cortex GABA levels[77] and we have reported direct experimental proof in adults, that visual processing differences in ASD are GABA-dependent[78].

Some strengths and limitations of our study should be mentioned. We studied state-of-the-art infant fMRI acquired with a dedicated neonatal multiband EPI pipeline which features high temporal resolution (TR = 392 ms) but acknowledge this has relatively low signal-to-noise ratio in the deep grey matter structures, including thalamus, basal ganglia, and brainstem[79]. We chose a LEiDA processing pipeline, as it provides time-resolved metrics of brain states and has been successfully applied to study brain dynamics in adults[28,29]. A majority of prior studies of dynamic brain functional connectivity have adopted sliding window approaches, which necessitates the choice of arbitrary parameters in the analysis, such as window and step sizes. There is little consensus on these parameters as the use of small windows can magnify spurious variations and large windows can soften sharper changes in brain dynamics[80–82]. Thus, the choice of a time-resolved approach like LEiDA is a strength in our methodology, though other techniques such as Hidden Markov Models are also available[83]. Our analysis treated brain state occurrence in an independent fashion, i.e., without memory. Future studies could benefit from developing a metric that considers how brain states are impacted by previously occurring states. Our results are also potentially limited by the use of the AAL atlas to define our regions of interest. While this atlas has been adapted for neonatal use[32], structurally defined atlases could potentially poorly correlate with functional boundaries in the brain, particularly during the neonatal period[2,84]. A possible solution for future work could be the adaptation and use of multi-modal generated surface atlases for neonatal fMRI studies[85]. However, we reproduced our pipeline using an alternative parcellation scheme (the M-CRIB atlas),

obtaining similar results to those featured by the neonatal AAL atlas. The dHCP cohort focused on characterising typical development, hence this study features an unbalanced number of term vs preterm infants. Preterm-born babies included were predominantly moderate or late preterm, and mostly "healthy", with no incidental findings of clinical significance. While this may be better representative of general preterm population (80% of preterm infants are born moderate or late preterm[86]) we cannot extrapolate our results to very or extremely preterm born babies, or to those with significant white matter damage. In our analysis, we also did not consider the association with socio-demographic factors and social deprivation in early neurodevelopmental outcomes beyond using the Index of Multiple Deprivation scores as a covariate. However, multiples studies have shown that family psychosocial and socio-demographic factors have a significant impact on brain development in childhood; with factors such as maternal stress, depression, low education, maternal immigration status, maternal age greater than 35 years, paternal age over 38 years and low household income all being linked to poorer developmental outcomes[87–91]. Thus, further studies could benefit from evaluating links between these wider socio-demographic markers and brain dynamics in early childhood[92].

In conclusion, in this study we evaluated global functional brain dynamics and transient brain states in the newborn brain. Our approach allowed us to define a set of six fundamental transient brain states, which are comprised by structures previously shown to be established in earlier phases of brain development. We have highlighted the impact brain maturation has on brain dynamics, as well as atypical patterns associated with preterm birth. Brain state dynamics at birth appear to be functionally relevant as they are correlated with a range of neurodevelopmental outcomes in early childhood. This encourages further work to understand their prognostic value and regulation to guide support and intervention where appropriate.

## Methods
### Participants
We analysed a total of 390 (out of 809) datasets from the dHCP (release 3)[23] acquired from both term ($n = 324$) and preterm born (gestational age (GA) at birth <37 weeks; $n = 66$) babies. Full inclusion and exclusion criteria and sample excluded at each step are detailed in Supplementary Figure S8. Term-born babies were born at median GA at birth of 40.14 weeks (IQR = 1.86 weeks) and scanned soon after birth (median postmenstrual age (PMA) at scan = 41.57, IQR = 2.43 weeks). Preterm-born babies were born at a median GA at birth of 33.36 weeks (IQR = 5.86 weeks) and scanned at term-equivalent age (median PMA at scan = 40.5 (IQR = 2.71)). Table 2 shows demographic data of the sample. The distribution of PMA at scan and GA at birth for the individuals included in this study is shown in Supplementary Fig. S9. All children were invited to the Centre for the Developing Brain, St Thomas' Hospital, London, for neurodevelopmental evaluation by experienced paediatricians or psychologists at 18 months after expected delivery date. The Bayley Scales of Infant and Toddler Development, Third Edition (Bayley-III)[30] were used to assess general developmental outcomes across motor, language and cognitive domains in 305 individuals from the total population, comprising 257 infants born at term and 48 born preterm (higher scores indicate greater skills). The Quantitative Checklist for Autism in Toddlers (Q-CHAT)[31] at 18 months corrected age was available in 300 individuals (254 born at term and 46 born preterm), as a measure of atypical social, sensory and repetitive behaviours which occur as a continuum in the population[31]. Although higher Q-CHAT scores may indicate more threshold or subthreshold autistic traits, we emphasise our use of this instrument was to capture behaviours not tapped by the BSID-III, rather than to screen for ASD. The index of multiple deprivation (IMD) rank – a composite measure of geographical deprivation estimated

from the address of the mother at the time of birth[93] – was obtained for every subject and included as a covariate for models aimed at assessing the relationship between neonatal brain dynamics and subsequent neurodevelopmental and behavioural outcomes.

## MRI acquisition

We evaluated fMRI scans obtained as part of the dHCP at the Evelina Newborn Imaging Centre, Evelina London Children's Hospital, using a 3 Tesla Philips Achieva system (Philips Medical Systems). Ethical approval was given by the UK National Research Ethics Authority (14/LO/1169), and written consent was obtained from all participating families prior to data collection. Scans were performed without sedation in a dedicated neonatal set-up with optimised transport system, positioning devices, hearing protection, and custom-built 32-channel receive head coil and acoustic hood[94]. Scans were supervised by a neonatal nurse or paediatrician who monitored heart rate, oxygen saturation and temperature throughout the duration of the scan. Blood-oxygen-level-dependent (BOLD) fMRI was acquired using a multi-slice echo planar imaging sequence with multiband excitation (factor 9) (repetition time (TR) = 392 ms, echo time (TE) = 38 ms, voxel size = $2.15 \times 2.15 \times 2.15$ mm, flip angle = 34°, 45 slices, total time = 15 m 3 s, number of volumes = 2300)[23]. Anatomical images were acquired for brain morphometry and clinical reporting[23]. T1-weighted images had a reconstructed spatial resolution = $0.8 \times 0.8 \times 0.8$ mm, field of view = $145 \times 122 \times 100$ mm, TR = 4795 ms. T2-weighted images had a reconstructed spatial resolution = $0.8 \times 0.8 \times 0.8$ mm, field of view = $145 \times 145 \times 108$ mm, TR = 12 s, TE = 156 ms. fMRI datasets with excessive motion (more than 10% of motion outliers[19]), or incidental MRI findings of clinical significance (radiology scores 4 or 5 which indicate major lesions within white matter, cortex, basal ganglia or cerebellum—as described in dHCP database[23]) were excluded. In cases of twin/triplet scans only one infant was included (the one with least motion outliers during acquisition).

## Data processing

Individual fMRI datasets were pre-processed according to the dHCP dedicated neonatal pipeline[79]. Briefly, local distortion due to field inhomogeneity was corrected using *topup*[95]; intra- and inter-volume motion correction; and associated dynamic distortions correction using rigid-body realignment and slice-to-volume *eddy*[96]. Residual motion, multiband acquisition, and cardiorespiratory artefacts were regressed out using FSL FIX[97].

Head motion has been shown to produce spurious and systematic correlations in resting state fMRI[98]. In addition to specific processing steps within the dHCP pipeline implemented to minimise motion artefact on the BOLD signal[79], we also evaluated motion magnitude for each dataset using framewise displacement (FD)[98]. To minimise the effects of motion (and/or any likely associated differences) on the determination of brain states, only participants with less than 10% motion outliers (defined as FD > 75th centile + 1.5*IQR) were selected for the final analysed subsample. The total number of motion outliers was additionally used as a covariate of control in the analysis models described in the statistics section.

We segmented the T2-weighted volumes into nine tissue types including white matter, grey matter, and cerebrospinal fluid with a dedicated neonatal tissue segmentation pipeline[99]. We parcellated each subject's T2-weighted volume in 90 cortical and subcortical parcels using the Anatomical Automated Labels (AAL) atlas[100], mapped to the neonatal brain[32], adapted and manually corrected into the dHCP high-resolution template[101]. We transformed the AAL atlas from template space into each subject's native space with a non-linear registration based on a diffeomorphic symmetric image normalisation method (SyN)[102] using T2-weighted contrast and the segmentation obtained previously. Grey matter segmentation and parcels were propagated from T2-weighted native space into each subject's fMRI space with a boundary-based linear registration available as part of the functional dHCP processing pipeline[79]. Average BOLD timeseries were then calculated for each of the 90 AAL parcels in their intersection with grey matter, deep grey matter, or basal ganglia segmentation masks, as appropriate. An alternative parcellation scheme was also used, following the same procedure but using 80 cortical regions from the M-CRIB atlas[103], a neonatal adaptation of the Desikan-Killiany atlas[104].

## Analysing BOLD timeseries

We filtered the BOLD timeseries with a bandpass Butterworth second order filter in the range of 0.02–0.10 Hz[29] and obtained the phases $\varphi_j(t)$ for each parcel in time with the Hilbert transform. For a given real

**Table 2 | Demographic details of the term- and preterm-born groups**

| | Term (n = 324) | Preterm (n = 66) | Statistic | p value |
|---|---|---|---|---|
| Demographics & Clinical Details | | | | |
| GA at birth [in weeks], Mean (SD, range) | 40.04 (1.26, 37.00–42.71) | 32.79 (3.40, 23.71–36.86) | 12.814[a] | p < 0.001 |
| PMA at scan [in weeks], Mean (SD, range) | 41.60 (1.67, 37.43–44.86) | 40.60 (2.13, 37.00–45.14) | 3.615[a] | p < 0.001 |
| PND at scan [in weeks], Mean (SD, range) | 1.56 (1.34, 0.00–7.00) | 7.81 (4.62, 0.14–19.72) | −9.857[a] | p < 0.001 |
| IMD[e], Mean (SD, range) | 13892.19 (7481.54, 614–32731) | 17960.02 (7679.92, 3393–31478) | −3.182[a] | p < 0.001 |
| Sex [female count] (%) | 149 (45.99%) | 26 (39.39%) | 0.716[b] | p = 0.398 |
| % FD outliers (S.D.) | 5.38 (2.62) | 4.52 (2.49) | 2.451[a] | p = 0.014 |
| Follow-up | | | | |
| Corrected age at follow-up[a] [months], Mean (SD, range) | 18.83 (1.30, 17.27–24.33) | 18.71 (1.35, 17.47–23.87) | 0.672[a] | p = 0.503 |
| Uncorrected age at follow-up[a] [months], Mean (SD, range) | 18.82 (1.34, 16.97–24.47) | 20.40 (1.45, 18.30–24.87) | −7.060[a] | p < 0.001 |
| Bayley III[c] - cognitive (S.D.) | 101.56 (10.70) | 99.90 (13.11) | 0.618[a] | p = 0.538 |
| Bayley III[c] - motor (S.D.) | 102.25 (9.72) | 99.65 (10.26) | 1.733[a] | p = 0.083 |
| Bayley III[c] - language (S.D.) | 99.33 (15.72) | 95.69 (15.85) | 1.155[a] | p = 0.249 |
| Q-CHAT[d] (S.D.) | 29.90 (8.51) | 31.63 (11.81) | −0.682[a] | p = 0.497 |

*FD* framewise displacement.
[a]Z (Mann-Whitney *U* test).
[b]$\chi^2$-test.
[c]Bayley Scales of Infant Development: Third Edition (Bayley-III) - # of complete assessments: 257 term, 48 preterm.
[d]Quantitative Checklist for Autism in Toddlers (Q-CHAT) - # of complete assessments: 254 term, 46 preterm.
[e]Index of Multiple Deprivation - # of complete assessments: 247 term, 43 preterm.

signal $s(t)$, we built a complex signal $z(t)$[33,105] given by:

$$z(t) = s(t) + iH[s(t)] \qquad (1)$$

In which $H[s(t)]$ represented a Hilbert transform applied to the real signal s(t) and is defined below, with p.v. consisting of Cauchy principal value[105]:

$$H[s(t)] = p.v. \int_{-\infty}^{\infty} \frac{s(t-\tau)}{\pi t} dt \qquad (2)$$

The phases $\varphi_j(t)$ for each parcel can be calculated directly from z(t):

$$\varphi(t) = \arctan\left(\frac{H[s(t)]}{s(t)}\right) \qquad (3)$$

### Kuramoto order parameter

The Kuramoto Order Parameter (KOP) measures the global level of synchronicity of multiple oscillators and is defined in equation below, where $\varphi j(t)$ is the signal phase of an oscillator $j$ at a given time. In our study, each brain parcel (AAL region) is treated as an independent oscillator.

$$KOP(t) = \frac{1}{N}\left|\sum_{\forall j=1}^{N} e^{i\varphi_j(t)}\right| \qquad (4)$$

Once KOP was obtained for every time point, we obtained the mean KOP (synchrony aggregate over time, referred as "mean synchronisation"), KOP standard deviation (referred as "metastability")[25,26]. Mean KOP provides a broad measure of whole brain synchronicity, whereas metastability provides a measure of how synchronisation between different oscillators fluctuates over time, i.e., brain flexibility[11,25,27].

### The Leading Eigenvector Analysis (LEiDA)

KOP analyses can provide insight on global dynamic properties over all oscillators (brain parcels); but cannot inform which specific brain structures might be involved in those changes. To evaluate such modular (local) properties we applied the LEiDA approach—which allowed us to investigate phase coherence in different sets of parcels. To do so we first calculated the phase difference between a parcel $i$ and a parcel $j$ at every instant (TR), using the cosine distance:

$$\Delta\varphi_{ij}(t) = \cos\left(\varphi_j(t) - \varphi_i(t)\right) \qquad (5)$$

This results in a symmetric dynamic functional connectivity matrix for each fMRI volume. We then obtain a lower-dimensional representation with the LEiDA approach[28,29], whereby the LEiDA vector corresponds to the first eigenvector of the decomposition of the matrix $\Delta\varphi_{ij}(t)$. This method has been previously shown to reveal information on the community structure of networks and graphs[106]. Once the LEiDA vectors were obtained, we clustered them using K-Means[28,29] with the optimal $K$ (six) determined heuristically with the Calinski-Harabasz and Davies-Bouldin methods (Supplementary Figure S10.

Each cluster represents a set of LEiDAs, and we refer to each of these as a *brain state*. The dynamics of such states can be studied with three main metrics: fractional occupancy—which refers to the total proportion of time spent in a given state or probability of that state; dwell time—which consists of the average continuous time spent on each state; and Markovian probabilities of transitions between each state[28,29]. In addition, we also calculated values for mean

synchronisation and metastability for each state by averaging those for the volumes belonging to each cluster.

### Statistics

Firstly, we restricted our sample to the term-born individuals only ($n = 324$) and evaluated the association with brain maturation and ex-utero experience (with PMA and PND at scan, respectively) in global and modular brain dynamics. Secondly, we evaluated the association brain dynamics with preterm birth by studying the entire sample of 390 individuals. Prematurity was coded as a binary variable with 1 for preterm-born individuals (GA at birth less than 37 weeks) and 0 for term-born participants (GA at birth of 37 weeks or more). Finally, we evaluated the association of global and modular dynamic features with later neurodevelopmental outcomes at 18 months corrected age ($n = 305$ – Bayley-III; $n = 300$ – Q-CHAT).

### Global dynamics

We characterised the association with age and postnatal experience (PMA and PND at scan) by fitting the linear model GLM1 (324 term-born babies): $y \sim \beta_0 + \beta_1 PMA + \beta_2 PND + \beta_3 Sex + \beta_4 [Motion\ outliers\ (FD)]$. To characterise the association with preterm birth on brain dynamics, we fitted the linear model GLM2 (324 term-born and 66 preterm-born babies): $y \sim \beta_0 + \beta_1 Preterm\text{-}born + \beta_2 PMA + \beta_3 Sex + \beta_4 [Motion\ outliers\ (FD)]$. We assessed the association of brain global dynamics with cognitive and behavioural outcome measures, i.e., Bayley-III and Q-CHAT, at 18 months in a model given by GLM3 (257 term-born and 48 preterm-born babies for Bayley-III scores; and 254 term-born and 46 preterm-born babies for Q-CHAT): $y \sim \beta_0 + \beta_1 GA + \beta_2 PMA + \beta_3 Sex + \beta_4 [Motion\ outliers\ (FD)] + \beta_5 [Corrected\ age\ at\ assessment] + \beta_6 [Assessed\ Component] + \beta_7 [Index\ of\ Multiple\ Deprivation]$ (with *Assessed Component* consisting of Bayley's cognitive, Bayley's language, Bayley's motor, or Q-CHAT total scores).

### Modular dynamics: brain states

Firstly, we tested differences between the brain states defined in this study in terms of their mean synchronisation, metastability, fractional occupancy, and dwell times per subject via a type III ANOVA with Satterthwaite's method[107] and the linear mixed effects model GLME1 (including 324 term-born): $y \sim \beta_0 + \beta_1 State + (1 | Subject\ ID)$ – with Subject ID accounting for the random effect. By fitting GLM1 (324 term-born babies): $y \sim \beta_0 + \beta_1 PMA + \beta_2 PND + \beta_3 Sex + \beta_4 [Motion\ outliers\ (FD)]$, we characterised the association with age (PMA at scan) and postnatal experience (PND at scan) on fractional occupancy, dwell times, mean synchronisation, and metastability for each of the six brain states.

Secondly, to characterise the association with preterm birth on brain states and state-change probabilities, we fitted GLM2 (324 term-born and 66 preterm-born babies): $y \sim \beta_0 + \beta_1 Preterm\text{-}born + \beta_2 PMA + \beta_3 Sex + \beta_4 [Motion\ outliers\ (FD)]$ and GLM3 (324 term-born and 66 preterm-born babies): $y \sim \beta_0 + \beta_1 GA + \beta_2 PMA + \beta_3 Sex + \beta_4 [Motion\ outliers\ (FD)]$.

Thirdly, we assessed the association of brain dynamics with neurodevelopmental outcome measures (i.e., Bayley-III and Q-CHAT at 18 months) in a model given by GLM4 (257 term-born and 48 preterm-born babies for Bayley-III; and 254 term-born and 46 preterm-born babies for Q-CHAT): $y \sim \beta_0 + \beta_1 GA + \beta_2 PMA + \beta_3 Sex + \beta_4 [Motion\ outliers\ (FD)] + \beta_5 [Corrected\ age\ at\ assessment] + \beta_6 [Assessed\ component] + \beta_7 [Index\ of\ Multiple\ Deprivation]$ (with *Assessed Component* consisting of Bayley's cognitive, Bayley's language, Bayley's motor, or Q-CHAT total scores).

### Statistical significance and repeated measures

We evaluated the statistical significance of each variable of interest with two-sided permutation tests with 10,000 repetitions for all GLMs. P-values are reported uncorrected, highlighting those surviving

multiple comparison correction across states using Benjamini-Hochberg FDR method with $\alpha$ error at 5%[108].

### Reporting summary

Further information on research design is available in the Nature Portfolio Reporting Summary linked to this article.

## Data availability

The AAL-UNC atlas adapted to the dHCP template space, and pre-processed BOLD timeseries data generated in this study have been deposited in the Zenodo database under accession code 10.5281/zenodo.7053984. The fMRI datasets and clinical data are available under restricted access as per dHCP data release conditions, access can be obtained via https://data.developingconnectome.org. Source data are provided with this paper.

## Code availability

Dynamic properties of the BOLD signal fluctuations were assessed with dynFC: CoDe-Neuro's Dynamic Functional Connectivity Tools[109], a set of scripts written in Python v3.7 (https://code-neuro.github.io/dynfc/), and supporting libraries *Numpy, SciPy, Scikit-learn, pickle, h5py, pandas, os, sys,* and *feather.*

Statistics and figures were produced in R programming language and auxiliary packages *ggplot2, tidyr, dplyr, cowplot, purrr, RColorBrewer, knitr, janitor, ggExtra, stringr, rjson, broom, Tidymodels, coin, shadowtext, effsize,* modelr, ggimage, ggpubr, *patchwork, ggbeeswarm, ggrepel, ggtext, MetBrewer, lmerTest, forcats, stateR, p-testR, broom.mixed, lme4,* and *DiagrammeR.*

Brain volume images were produced with BrainNet Viewer and Tools for NIfTI and ANALYZE images. All scripts used in this article's statistics and figures; and relevant instructions on how to run them, are available in https://github.com/CoDe-Neuro/neonatal_dfc[109,110].

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

## Acknowledgements

This work was supported by the European Research Council under the European Union's Seventh Framework Programme (FP7/20072013)/ERC grant agreement no. 319456 (dHCP project) and a Wellcome Trust Seed Award in Science [217316/Z/19/Z] to DB. This study represents independent research part funded by the National Institute for Health and Care Research (NIHR) Maudsley Biomedical Research Centre (BRC) at South London and Maudsley NHS Foundation Trust and King's College London. The authors also acknowledge support in part from the Engineering and Physical Sciences Research Council (EPSRC) Centre for Medical Engineering at Kings College London [WT 203148/Z/16/Z], MRC strategic grant [MR/K006355/1], the Department of Health through an NIHR Comprehensive Biomedical Research Centre Award (to King's College Hospital NHS Foundation Trust). The results leading to this publication have received funding from the Innovative Medicines Initiative 2 Joint Undertaking under grant agreement No 777394 for the project AIMS-2-TRIALS. This Joint Undertaking receives support from the European Union's Horizon 2020 research and innovation programme and EFPIA and AUTISM SPEAKS, Autistica, SFARI. S.F.M. and O.G.G. were supported by grants from the UK Medical Research Council [MR/N013700/1] and [MR/P502108/1] respectively. J.O.M., T.A., G.M., and A.D.E. received support from the Medical Research Council Centre for Neurodevelopmental Disorders, King's College London [MR/N026063/1]. L.C.-G. received support from Project PID2021–129022OA-I00 funded by MCIN/AEI/10.13039/501100011033/FEDER, EU, and CAM-Spain under the line support for R&D projects for Beatriz Galindo researchers BGP18/00178. J.J.T. was supported by grants from the Finnish Medical Foundation, Sigrid Juselius Foundation, Signe and Ane Gyllenberg Foundation, Emil Aaltonen Foundation and Hospital District of Southwest Finland State Research Grants. J.O.M. is supported by a Sir Henry Dale Fellowship jointly funded by the Wellcome Trust and the Royal

Society [206675/Z/17/Z]. T.A. is supported by an MRC Clinician Scientist Fellowship [MR/P008712/1] and Transition Support Award [MR/V036874/1]. The authors acknowledge use of the research computing facility at King's College London, Rosalind (https://rosalind.kcl.ac.uk), which is delivered in partnership with the National Institute for Health Research (NIHR) Biomedical Research Centres at South London & Maudsley and Guy's & St. Thomas' NHS Foundation Trusts, and part-funded by capital equipment grants from the Maudsley Charity (award 980) and Guy's & St. Thomas' Charity (TR130505). The views expressed are those of the authors and not necessarily those of the funders, the NHS, the National Institute for Health Research, the Department of Health and Social Care, or the IHI-JU2. The funders had no role in the design and conduct of the study; collection, management, analysis, and interpretation of the data; preparation, review, or approval of the manuscript; and decision to submit the manuscript for publication.

## Author contributions

Conceptualisation: L.G.S.F., D.B.; methodology: L.G.S.F., S.F.G., L.C.G., A.N.P., G.D., J.V.H., D.B.; data analysis: L.G.S.F., D.B.; data acquisition: J.C., S.F., E.H., R.D., J.V.H., A.N.P., A.C., T.A.; interpretation of results: L.G.S.F., J.C., O.G.G., S.F.M., E.D., J.J.T., G.D., S.J.C., J.O.M., A.C., C.N., T.A., A.D.E., G.M.A., D.B.; funding acquisition: D.B., G.M.A., A.D.E.; project administration: D.B., G.M.A., A.D.E.; supervision: D.B., G.M.A., A.D.E.; writing—original draft: L.G.S.F., J.C., D.B.; writing—review & editing: All authors.

## Competing interests

The authors report no competing interests.

## Additional information

[1]Department of Forensic and Neurodevelopmental Science, Institute of Psychiatry, Psychology & Neuroscience, King's College London, London SE5 8AF, UK. [2]Centre for the Developing Brain, School of Biomedical Engineering & Imaging Sciences, King's College London, London SE1 7EH, UK. [3]Department of Computer and Information Sciences, Faculty of Engineering and Environment, Northumbria University, Newcastle upon Tyne NE1 8ST, UK. [4]Oxford Centre for Functional Magnetic Resonance Imaging of the Brain, Wellcome Centre for Integrative Neuroimaging, University of Oxford, Oxford OX3 9DU, UK. [5]Biomedical Image Technologies, ETSI Telecomunicación, Universidad Politécnica de Madrid, 28040 Madrid, Spain. [6]Centro de Investigación Biomédica en Red de Bioingeniería, Biomateriales y Nanomedicina, Instituto de Salud Carlos III, 28029 Madrid, Spain. [7]MRC Centre for Neurodevelopmental Disorders, King's College London, London SE1 1UL, UK. [8]Department of Brain Sciences, Imperial College London, London W12 0BZ, UK. [9]UK Dementia Research Institute at Imperial College London, London W12 0BZ, UK. [10]FinnBrain Birth Cohort Study, Turku Brain and Mind Center, Institute of Clinical Medicine, University of Turku, 20500 Turku, Finland. [11]Turku Collegium for Science and Medicine and Technology, University of Turku, 20500 Turku, Finland. [12]Department of Psychiatry, University of Turku and Turku University Hospital, 20500 Turku, Finland. [13]Center for Brain and Cognition, Computational Neuroscience Group, Department of Information and Communication Technologies, Pompeu Fabra University, 08002 Barcelona, Spain. [14]Catalan Institution for Research and Advanced Studies, 08010 Barcelona, Spain. [15]Department of Neuropsychology, Max Planck Institute for Human Cognitive and Brain Sciences, 04103 Leipzig, Germany. [16]School of Psychological Sciences, Monash University, Melbourne, VIC 3010, Australia. [17]Department of Child and Adolescent Psychiatry, Institute of Psychiatry, Psychology & Neuroscience, King's College London, London SE5 8AF, UK. [18]Department of Paediatric Neurosciences, Evelina London Children's Hospital, Guy's and St Thomas' NHS Foundation Trust, London SE1 7EH, UK. [19]Department of Bioengineering, Imperial College London, London SW7 2AZ, UK. [20]These authors contributed equally: A. David Edwards, Grainne McAlonan, Dafnis Batalle. ✉e-mail: dafnis.batalle@kcl.ac.uk

