## [Peer Review File · Nature Communications]

Neonatal brain dynamic functional connectivity in term and preterm infants and its association with early childhood neurodevelopmentReviewer #1 (Remarks to the Author):

This manuscript is looking at dynamic functional connectivity using fMRI, recorded at term age or term equivalent age in preterm-born neonates. Thus, they have a single fMRI study in the infants, all at approximately the same age (corrected age for the preterms). The dynamic aspect is novel, but many other studies (using a range of structural brain metrics and functional metrics) have shown that preterm birth impacts brain and brain function, that brain function changes with age and that brain function predicts outcomes. None of this is new – what could be interesting is the use of the dynamic analyses.

I have a number of suggestions and comments.

First, the number of infants in the two groups is very unbalanced. This is unusual for the group from King's College, where they are known to be leaders in studying the preterm brain and have large cohorts. Having only 66 preterms (and only 46 at 18 months follow-up) compared to the 324 term-born infants (254 at 18 months) certainly weakens the study. The GA at birth (range, not just mean) of the preterms needs to be given – as the mean is 33wks, it appears that they are mostly late preterm. This should be discussed.

I wonder why they chose the AAL atlas. There is a considerable literature showing that the AAL is very poor for resting-state fMRI and structural connectivity (e.g., Luppi et al. 2021, <https://direct.mit.edu/netn/article/5/1/96/97529/Combining-network-topology-and-information-theory>) - the poorest out of all atlases examined. Also, why would they choose to use an atlas based on structural parcellations instead of a functionally-derived atlas that is meant to be used for resting-state networks? They do mention these issues as limitations; I think better justification is needed.

Neither the technique (fMRI) nor the sample sizes are in the abstract. The authors need to give the age range for all the ages used, not just the mean age.

For the outcomes they have grouped the preterms and the terms together. I suspect that the preterms are driving the finding of a significant effect, as the preterm-born would be more likely to have higher Q-CHAT scores, and they also had the lower mean synchronization. Does this effect hold only for the term-born?

When they are correcting for multiple comparisons, what are they correcting over? This is not stated and should be.

I wonder about the meaningfulness of the data in Figure 1C - an R2 value of 0.02 is very low. Even though it passed significance, how is this meaningful?

How do they justify saying that the global features remained relatively stable through (should be 'in') early development when they only have a single scan at term age? (lines 315-316). Perhaps the age range is larger than one realises; again, ranges need to be presented, not just mean ages.

I am not sure that the argument that infants at term age, but with the longer postnatal age, are relying predominantly on higher order frontal parietal functions (lines 393-395) would hold with developmental psychologists.

The statement: 'Our results raise the possibility, that alterations in brain functional connectivity that are present soon after birth have functional consequences.' seems frankly trite – of course brain function has functional consequences.

As none of the cohort at 18 months had a diagnosis of autism, and there would be no indication as to how many might eventually have that diagnosis, I think the whole discussion of autism is beyond the data and should be deleted.

There are many instances when the language use is awkward – surprising given the number of native English speaker co-authors. Some examples (there are others):

- Line 51 – mental ill-health
- Line 117 – brain dynamics develop quickly with age – ‘at scan’ is irrelevant.
- Line 187 and forward – dwelling time – do they mean dwell time?
- similarly dwelling transitions? dwelling sequences??
- Many times, they say ‘later in childhood’, when they are referring to 18months – this is hardly later in childhood, it is barely out of infancy. Otherwise, it implies much longer follow-up.
- Line 383 - This chimes with other....
- Lines 433-434 – meaning is not clear at all to me.

Reviewer #2 (Remarks to the Author):

The authors investigated the characteristics of dynamic functional connectivity in a cohort of term-born (n=324) and preterm-born (n=66) neonates. They identified six fundamental transient states of neonatal dynamic functional connectivity and found disrupted brain dynamics in preterm-born neonates. They also found that brain state dynamics at birth appear to be correlated with neurodevelopment outcomes in early childhood. Overall, there is much to applaud in this manuscript. The authors firstly characterize the fundamental properties in early human development. Their findings would guide future work to explore prognostic value and intervention of fundamental dynamic characteristics of early human development. Some questions and suggestions which may improve the manuscript are provided below:

ABSTRACT, line 64, where is the difference between synchronisation and synchrony? Like line168, will using regional constrained states make it clear for readers?

Introduction, line 101, what is the motivation for using Kuramoto Order Parameter (KOP) and Leading Eigenvector Analysis (LEiDA) in this study? What is the advantage of KOP and LEiDA? Why KOP and LEiDA are the state-of art techniques? Although authors suggested the limitations of studies based on sliding window approaches, like hidden (Semi-)Markov models also could be employed this problem.

Results, line 197, “....., with probabilities above 83% for all six states”, however, in Figure 3A, the minimum dwelling transitions probability is 89.75%.

Results, line 198-201, why did 12 most frequent transitions incorporate an indirect transition to and from the high whole-brain synchronization global state A instead of state FP? As for state FP, these transitions also occurred via the intermediate states, B and C.

Results, line 231-233, “.....transitions from intermediate whole brain synchronisation state B to the occipital state increased with PMA at scan but decreased with PND at scan.”, however, in Figure3E, F and Figure 4H, there are no transitions from intermediate whole brain synchronisation state B to the state occipital.

Methods, Global dynamics, line 641, how to define Preterm-born for term-born? Why did authors use PND as dependent variable in GLM1 but not in GLM2?

In Figure 3C, D, Figure 4E, and Figure 5, authors are need to add the meaning of values in circle to the illustration of figure.

Reviewer #1

This manuscript is looking at dynamic functional connectivity using fMRI, recorded at term age or term equivalent age in preterm-born neonates. Thus, they have a single fMRI study in the infants, all at approximately the same age (corrected age for the preterms). The dynamic aspect is novel, but many other studies (using a range of structural brain metrics and functional metrics) have shown that preterm birth impacts brain and brain function, that brain function changes with age and that brain function predicts outcomes. None of this is new – what could be interesting is the use of the dynamic analyses.

I have a number of suggestions and comments.

We are grateful for the reviewer's fair assessment of our manuscript, and detailed suggestions.

- [x] First, the number of infants in the two groups is very unbalanced. This is unusual for the group from King's College, where they are known to be leaders in studying the preterm brain and have large cohorts. Having only 66 preterms (and only 46 at 18months follow-up) compared to the 324 term-born infants (254 at 18 months) certainly weakens the study. The GA at birth (range, not just mean) of the preterms needs to be given – as the mean is 33wks, it appears that they are mostly late preterm. This should be discussed.

The reviewer raises an important point, and we thank them for their comment. However, the difference in numbers arises from a study design choice. As the goal of the overall dHCP and this study was to map the structural and functional organisation of the developing brain, infants with severe or focal brain injuries were intentionally not included in the study population as this would significantly confound this mapping and limit interpretation. As such lesions are far more common in preterm infants born at very early gestations, the study population for the dHCP and this study inevitably had to consist of predominately moderate to late preterm infants, and relatively healthy. We have added the range of ages at birth for the participants included in our study in Table 2, as requested. We have commented on this matter in strengths and limitations section of the revised manuscript of our manuscript:

“The dHCP cohort focused on characterising typical development, hence this study features an unbalanced number of term vs preterm infants. Preterm-born babies included were predominantly moderate or late preterm, and mostly “healthy”, with no incidental findings of clinical significance. While this may be better representative of general preterm population (80% of preterm infants are born moderate or later preterm⁸³) we cannot extrapolate our results to very or extremely preterm born babies, or to those with significant white matter damage.”

- [x] I wonder why they chose the AAL atlas. There is a considerable literature showing that the AAL is very poor for resting-state fMRI and structural connectivity (e.g., Luppi et al. 2021, [<https://direct.mit.edu/netn/article/5/1/96/97529/Combining-network-topology-and-information-theory>] - the poorest out of all atlases examined. Also, why would they choose to use an atlas based on structural parcellations instead of a functionally-derived atlas that is meant to be used for resting-state networks? They do mention these issues as limitations; I think better justification is needed.

Thanks for raising this point. We chose the AAL atlas as it has been specifically adapted to the neonatal brain (Shi et al. 2011) and extensively used in the literature, which allows comparison with previous work by us and others (Batalle et al. 2017, Ciarrusta et al. 2020, Fenske et al. 2023). Importantly, functional connectivity is rapidly developing during the neonatal period (Eyre et al. 2021), and hence an atlas of functionally-derived regions from an adult population is unlikely to be more appropriate than an anatomical atlas specifically adapted to the neonatal brain.

Nonetheless, in line with reviewer’s point, we do think that it is fair and pertinent to ask whether our results are driven by the choice of a specific atlas. In order to show the resilience of our results to the atlas chosen, we have rerun all our analyses with an additional neonatal atlas, the M-CRIB atlas (Alexander et al. 2017, which is a well-known adaptation of the Desikan-Killiany atlas (Desikan et al. 2006) to the neonatal brain. The results of this analysis show that our results are robust to the choice of atlas as we found very similar results to those for the AAL atlas. We obtain similar global metrics results, i.e., synchronisation and metastability, and also show similar effects of preterm birth on global brain dynamics. We have also found regional brain states that are similar to those obtained with AAL atlas. Moreover, the associations with metrics, i.e., FO, DT, synchronisation, and metastability were all compatible with the results using the AAL parcellation. We also found associations between sensorimotor cortex metrics and outcome variables indicating changes in the same direction with mean synchronisation. We have added the new results for M-CRIB atlas as supplementary material, discussed the M-CRIB results in the revised manuscript and kept AAL in the main text.

The following paragraphs were added to the main text:

“Results were robust to the choice of atlas parcellation, as similar results were obtained with the Melbourne Children’s Regional Infant Brain (M-CRIB) atlas (Supplementary Table S1), including a significant association of metastability with PND at scan ($p = 0.019$), and reduced mean synchronisation and metastability in preterm-born infants when compared with term counterparts (Cohen’s $D = 0.628$, $p < 0.001$; and Cohen’s $D = 0.480$, $p < 0.001$ respectively, see Supplementary Table S1). Analogously, M-CRIB’s mean synchronisation was also significantly associated with Q-CHAT scores ($t = -2.7$, $p = 0.009$).”

Supplementary Table S1 | Association of global dynamic features (synchrony and metastability) with PMA and PND at scan, and effect of preterm birth using M-CRIB atlas.

	Term ($n = 324$)				Term vs Preterm ($n = 390$)			
	PMA at scan		PND at scan		Term ($n = 324$)	Preterm ($n = 66$)	Cohen’s D	p - value ^{††}
	t^{\dagger}	p -value [†]	t^{\dagger}	p -value [†]	[mean (S.D.)]			
Mean synchronisation	-0.173	0.862	1.370	0.172	0.53 (0.08)	0.48 (0.08)	0.628	$p < 0.001^*$
Metastability	0.753	0.448	-2.330	0.019*	0.20 (0.02)	0.19 (0.02)	0.480	$p < 0.001^*$

[†]GLM1 (including 324 term-born babies): $y \sim \beta_0 + \beta_1 PMA + \beta_2 PND + \beta_3 Sex + \beta_4 Motion$ outliers (FD). ^{††}GLM2 (including 324 term-born and 66 preterm-born babies): $y \sim \beta_0 +$

β_1 Preterm-born + β_2 PMA + β_3 Sex + β_4 Motion outliers (FD). * *p*-values surviving FDR multiple comparison correction with α error at 5%).

And we added a new sub-section to our revised Results section:

“A similar pattern of regional brain dynamics was obtained with an alternative parcellation scheme. The six brain states obtained with the alternative M-CRIB neonatal atlas were compatible with the ones obtained for the AAL atlas (Supplementary Figure S3), with three global synchronisation states and three more regionally constrained. States 1, 2 and 3 obtained with the M-CRIB atlas were consistent with global synchronisation states obtained with the AAL atlas. The M-CRIB’s State 4 featured some of the structures present in occipital state, State 5 featured similar structures to the sensorimotor state, and State 6 was concordant with FP state obtained from AAL.

Moreover, for the analyses using the M-CRIB atlas in term-born babies (Supplementary Figure S4), PMA was also associated with increased fractional occupancy ($t = 2.9$; $p = 0.002$) and mean synchronisation ($t = 3.0$; $p = 0.003$) for State 5 (~SM); and PND was associated with reduced dwell times ($t = -2.5$; $p = 0.012$) for State 5 (~SM). Transitions from States 6 (~FP) to State 2 (~Glb.) also increased with PND ($t = 3.3$; $p = 0.002$). Pre-term birth was also associated with similar changes in state metrics for M-CRIB atlas (Supplementary Figure S5) with increased fractional occupancy for State 4 (~Occ.) ($t = 2.3$; $p = 0.020$) and State 6 (~FP) ($t = 3.0$; $p = 0.002$); and reduced mean synchronisation for State 6 (~FP) ($t = -2.2$; $p = 0.027$). See Supplementary Figure S6 for a similar analysis with GA.

Association with developmental outcomes in cognitive ($t = -4.0$; $p < 0.001$) and motor ($t = -3.4$; $p < 0.001$) components of Bayley-III showed negative associations with mean synchronisation for State 5 (~SM) and higher Q-CHAT scores were associated with reduced fractional occupancy for State 1 ($t = -2.6$; $p = 0.008$), see Supplementary Figure S7 for a summary. A detailed description of results using M-CRIB atlas for parcellation is available in Supplementary Materials.”

Supplementary Figure S3 | Brain states in neonates using M-CRIB atlas. (a) LEiDA vectors for each of the six brain states identified in the neonatal brain using M-CRIB parcels. **(b)** Representation of LEiDA on brain surfaces (right side view). **(c)** Representation of LEiDA on brain surfaces (left side view).

Supplementary Figure S4 | Brain dynamics in term-born children (n = 324) (M-CRIB atlas). (a) All transitions including dwelling state transitions. (b) Main transitions (top 12) between states excluding dwelling sequences. (c) Summary of brain state features significantly associated with PMA at scan. (d) Summary of brain state features significantly associated with PND at scan. (e) Summary of significant correlations between state transitions probabilities and PMA at scan. (f) Summary of significant correlations between state transitions probabilities and PND at scan. [†]GLM1 (including 324 term-born babies): $y \sim \beta_0 + \beta_1 PMA + \beta_2 PND + \beta_3 Sex + \beta_4 Motion\ outliers\ (FD)$. Values shown in c and d indicate t-statistics. All significant

associations shown in this figure survive FDR multiple comparison correction with α error at 5%.

Supplementary Figure S5 | Effect of preterm birth in brain dynamics (M-CRIB atlas). (a) Mean dwell times (DT). (b) Mean fractional occupancy (FO). (c) Mean synchronisation. (d) Metastability. (e) Summary of significant associations with preterm birth (f) Association of state transitions probabilities and preterm birth. †GLM1 (324 term-born babies): $y \sim \beta_0 + \beta_1 PMA + \beta_2 PND + \beta_3 Sex + \beta_4 Motion\ outliers (FD)$. ††GLM2 (324 term-born and 66 preterm-born babies): $y \sim \beta_0 + \beta_1 Preterm-born + \beta_2 PMA + \beta_3 Sex + \beta_4 Motion\ outliers (FD)$. * $p < 0.05$. ** $p < 0.01$. *** $p < 0.001$. Values shown in e indicate t-statistics. All significant associations highlighted survive FDR multiple comparison correction with α error at 5%.

Supplementary Figure S6 | Association of GA at birth with modular features of neonatal brain dynamics (M-CRIB atlas). (A) Summary of associations between each of the four metrics (dwell times, fractional occupancy, mean synchronisation, metastability) and GA at birth. (B) Association of state transitions probabilities and GA at birth. ^{†††}GLM3 (including 324 term-born and 66 preterm-born babies): $y \sim \beta_0 + \beta_1 GA + \beta_2 PMA + \beta_3 Sex + \beta_4 Motion$ outliers (FD). * $p < 0.05$. ** $p < 0.01$. *** $p < 0.001$. Occ.: Occipital. SM: Sensorimotor. FP: Frontoparietal.

Supplementary Figure S7 | Summary of associations of brain state features with neurodevelopmental outcomes at 18 months corrected age (M-CRIB atlas). Association of average (a) mean synchronisation and (b) fractional occupancy in each of the six defined brain states during perinatal period with cognitive (Cog. Comp), language (Lang. Comp.) and motor (Mot. Comp) Bayley-III composite scores and Q-CHAT scores. GLM3 (257 term-born and 48 preterm-born babies for Bayley-III; and 254 term-born and 46 preterm-born babies for Q-CHAT): $y \sim \beta_0 + \beta_1GA + \beta_2PMA + \beta_3Sex + \beta_4Motion\ outliers\ (FD) + \beta_5[Corrected\ age\ at\ assessment] + \beta_6[Assessed\ component] + \beta_7[Index\ of\ multiple\ deprivation]$. Values shown in in both panels indicate t-statistics. All significant associations highlighted survive FDR multiple comparison correction with α error at 5%.

- [x] Neither the technique (fMRI) nor the sample sizes are in the abstract.

Thank you for noting this, we have updated the abstract accordingly.

“In this study we characterised dynamic functional connectivity with functional magnetic resonance imaging (fMRI) in the first few weeks of postnatal life in term-born (n = 324) and preterm-born (n = 66) individuals”

- [x] The authors need to give the age range for all the ages used, not just the mean age.

We have updated the table to reflect the changes suggested by the reviewer. A figure featuring distributions is available in supplementary materials (Supplementary Figure S9).

Table 2. Demographic details of the term- and preterm-born groups.

	Term (n = 324)	Preterm (n = 66)	Statistic	p-value
Demographics & Clinical Details				
GA at birth [in weeks], Mean (SD, range)	40.04 (1.26, 37.00 – 42.71)	32.79 (3.40, 23.71 – 36.86)	12.814 ^a	$p < 0.001$
PMA at scan [in weeks], Mean (SD, range)	41.60 (1.67, 37.43 – 44.86)	40.60 (2.13, 37.00 – 45.14)	3.615 ^a	$p < 0.001$

PND at scan [in weeks], Mean (SD, range)	1.56 (1.34, 0.00 – 7.00)	7.81 (4.62, 0.14 – 19.72)	-9.857 ^a	$p < 0.001$
IMD ^{†††} , Mean (SD, range)	13892.19 (7481.54, 614 – 32731)	17960.02 (7679.92, 3393 – 31478)	-3.182 ^a	$p < 0.001$
Sex [female count] (%)	149 (45.99%)	26 (39.39%)	0.716 ^b	$p = 0.398$
% FD outliers (S.D.)	5.38 (2.62)	4.52 (2.49)	2.451 ^a	$p = 0.014$

Follow-up				
Corrected age at follow-up [†] [months], Mean (SD, range)	18.83 (1.30, 17.27 – 24.33)	18.71 (1.35, 17.47 – 23.87)	0.672 ^a	$p = 0.503$
Uncorrected age at follow-up [†] [months], Mean (SD, range)	18.82 (1.34, 16.97 – 24.47)	20.40 (1.45, 18.30 – 24.87)	-7.060 ^a	$p < 0.001$
Bayley III [†] - cognitive (S.D.)	101.56 (10.70)	99.90 (13.11)	0.618 ^a	$p = 0.538$
Bayley III [†] - motor (S.D.)	102.25 (9.72)	99.65 (10.26)	1.733 ^a	$p = 0.083$
Bayley III [†] - language (S.D.)	99.33 (15.72)	95.69 (15.85)	1.155 ^a	$p = 0.249$
Q-CHAT ^{††} (S.D.)	29.90 (8.51)	31.63 (11.81)	-0.682 ^a	$p = 0.497$

[†]Bayley Scales of Infant Development: Third Edition (Bayley-III)- # of complete assessments: 257 term, 48 preterm

^{††}Quantitative Checklist for Autism in Toddlers (Q-CHAT)- # of complete assessments: 254 term, 46 preterm

^{†††}Index of Multiple Deprivation - # of complete assessments: 247 term, 43 preterm

^aZ (Mann-Whitney U-test), ^b χ^2 -test

FD – Framewise Displacement

- [x] For the outcomes they have grouped the preterms and the terms together. I suspect that the preterms are driving the finding of a significant effect, as the preterm-born would be more likely to have higher Q-CHAT scores, and they also had the lower mean synchronization. Does this effect hold only for the term-born?

We have evaluated the reviewer's suspicion by fitting the same model for both term and preterm born data separately, as featured in the figure below.

When analysed separately, mean synchronisation in term born babies explain up to 2% of Q-CHAT variance, and up to 4% of Q-CHAT scores in preterm babies. However, only term-born participants had a statistically significant association ($p = 0.024$), suggesting that the results are not driven by preterm-born babies.

- [x] When they are correcting for multiple comparisons, what are they correcting over? This is not stated and should be.

Thank you for requesting this important clarification. We are correcting for multiple comparisons across states. We have amended the text describing this accordingly in the statistics section of the manuscript:

“P-values are reported uncorrected, highlighting those surviving multiple comparison correction across states using Benjamini-Hochberg False Discovery Rate (FDR) method with α error at 5%¹⁰⁵.”

- [x] I wonder about the meaningfulness of the data in Figure 1C - an R2 value of 0.02 is very low. Even though it passed significance, how is this meaningful?

We agree with the reviewer that this is indeed a small effect. The results of previous studies across the literature suggest the effects of various factors on the association of imaging at birth with outcome is complex and cumulative. Even factors widely accepted in the literature to affect this relationship such as GA at birth, which captures relevant clinical risk, explains just ~5% of the Q-CHAT variance at 18 months. Social deprivation, as measured by the index of multiple deprivation only explains up to ~10% of variance in neurodevelopmental outcomes (Gale-Grant et al. *bioRxiv*. 2022). In that context, we think it is worth reporting 2% of variance explained in typically developing children, particularly as it seems consistent with results obtained when assessing association of local dynamics, and robust to the atlas used (2% when using the MCRIB atlas). Notwithstanding, we agree with the reviewer that perhaps Figure 1C doesn't add much to our manuscript, so we decided to delete this panel from our revised manuscript.

Oliver Gale-Grant, Andrew Chew, Shona Falconer, Lucas G.S França, Sunniva Fenn-Moltu, Laila Hadaya, Nicholas Harper, Judit Ciarrusta, Tony Charman, Declan Murphy, Tomoki Arichi, Grainne McAlonan, Chiara Nosarti, A David Edwards, Dafnis Batalle.
bioRxiv 2022.09.26.508121; doi: <https://doi.org/10.1101/2022.09.26.508121>

- [x] How do they justify saying that the global features remained relatively stable through (should be ‘in’) early development when they only have a single scan at term age? (lines 315-316). Perhaps the age range is larger than one realises; again, ranges need to be presented, not just mean ages.

We apologise this wasn't clear in the original version of our manuscript. The data is of course cross-sectional, over a period of 7 weeks (37 to 44 weeks PMA at scan). We have now clarified this in the text and specified the range as requested:

*“We found that global dynamic features remained relatively stable **in** early **postnatal** development (37-44 weeks PMA at scan) in a term-born population, ...”*

- [x] I am not sure that the argument that infants at term age, but with the longer postnatal age, are relying predominantly on higher order frontal parietal functions (lines 393-395) would hold with developmental psychologists.

We have changed the text in this paragraph for clarity:

“Finally, we observed increased transitions from sensorimotor to frontoparietal structures with increasing duration of postnatal life, perhaps reflecting the emerging of functional maturation of frontoparietal systems which coincides with high interneuron migrations to these regions⁵², relative to the already mature sensorimotor systems⁴⁷.”

- [x] The statement: ‘Our results raise the possibility, that alterations in brain functional connectivity that are present soon after birth have functional consequences.’ seems frankly trite – of course brain function has functional consequences.

The reviewer is right, and we apologise for the trite remark. We have amended the manuscript:

“Our results raise the possibility, that alterations in brain dynamic functional connectivity that are present soon after birth have neurodevelopmental consequences later in life.”

- [x] As none of the cohort at 18 months had a diagnosis of autism, and there would be no indication as to how many might eventually have that diagnosis, I think the whole discussion of autism is beyond the data and should be deleted.

The reviewer is correct to state that emergence of autism traits in the general population is not equivalent to a clinical autism diagnosis, and we had tried to capture this in the discussion: *“our work did not examine predictors of a confirmed diagnosis of ASD”*. We emphasise that our study is not about diagnosed autism and completely agree that at 18 months no child has a diagnosis of autism. However, children who go on to receive a diagnosis of autism and those with a broader phenotype may already show emerging traits of the condition. This was the basis for development of the Q-CHAT (Allison et al., 2008, 2021). There is also accumulating evidence across modalities that infants who go on to receive a diagnosis of autism have differences in their neurobiology and physiology from as young as 6-9 months (Carter Leno et al. 2022; Fish et al. 2021; Haartsen et al. 2019; Kolesnik et al. 2019; Lloyd-Fox et al. 2018; Pote et al. 2019). Therefore, in exploring how dynamic functional connectivity in early life is associated to social cognitive development as assessed using the Q-CHAT, we do draw on the autism literature but have tried to frame this part of the discussion more clearly:

“We emphasise that we did not follow-up the children with higher Q-CHAT scores beyond 18 months, thus our work did not evaluate predictors of a confirmed diagnosis of ASD, nor is our study about diagnosed autism. However, children who go on to receive a diagnosis of autism and those with a broader phenotype may already show emerging traits of the condition. This was the basis for development of the Q-CHAT^{27,39}. There is also accumulating evidence across different modalities that infants who go on to receive a diagnosis of autism have differences in their neurobiology and physiology from as young as 6-9 months⁶²⁻⁶⁷. Thus our work links emerging social cognitive profile relevant to ASD, also to dynamic functional connectivity at birth, especially within sensory networks. This correlation between higher scores on an instrument, which captures early features relevant to ASD (though not necessary diagnostic) to the dynamics of sensory systems, is in agreement with the importance of sensory processes throughout the lifespan in individuals who have an ASD diagnosis.”

Allison, C. et al. The Q-CHAT (Quantitative CHECKlist for Autism in Toddlers): A Normally Distributed Quantitative Measure of Autistic Traits at 18–24 Months of Age: Preliminary Report. *J Autism Dev Disord* 38, 1414–1425 (2008).

Carter Leno, Virginia, Jannath Begum-Ali, Amy Goodwin, Luke Mason, Greg Pasco, Andrew Pickles, Shruti Garg, et al. 2022. 'Infant Excitation/Inhibition Balance Interacts with Executive Attention to Predict Autistic Traits in Childhood'. *Molecular Autism* 13 (1): 46. <https://doi.org/10.1186/s13229-022-00526-1>.

Fish, Laurel A., Pär Nyström, Teodora Gliga, Anna Gui, Jannath Begum Ali, Luke Mason, Shruti Garg, et al. 2021. 'Development of the Pupillary Light Reflex from 9 to 24 Months: Association with Common Autism Spectrum Disorder (ASD) Genetic Liability and 3-year ASD Diagnosis'. *Journal of Child Psychology and Psychiatry* 62 (11): 1308–19. <https://doi.org/10.1111/jcpp.13518>.

Haartsen, Rianne, Emily J. H. Jones, Elena V. Orekhova, Tony Charman, Mark H. Johnson, The BASIS team, S. Baron-Cohen, et al. 2019. 'Functional EEG Connectivity in Infants Associates with Later Restricted and Repetitive Behaviours in Autism; a Replication Study'. *Translational Psychiatry* 9 (1): 66. <https://doi.org/10.1038/s41398-019-0380-2>.

Kolesnik, Anna, Jannath Begum Ali, Teodora Gliga, Jeanne Guiraud, Tony Charman, Mark H. Johnson, Emily J. H. Jones, and The BASIS Team. 2019. 'Increased Cortical Reactivity to Repeated Tones at 8 Months in Infants with Later ASD'. *Translational Psychiatry* 9 (1): 46. <https://doi.org/10.1038/s41398-019-0393-x>.

Lloyd-Fox, S., A. Blasi, G. Pasco, T. Gliga, E. J. H. Jones, D. G. M. Murphy, C. E. Elwell, T. Charman, M. H. Johnson, and the BASIS Team. 2018. 'Cortical Responses before 6 Months of Life Associate with Later Autism'. *European Journal of Neuroscience* 47 (6): 736–49. <https://doi.org/10.1111/ejn.13757>.

Pote, Inês, Siying Wang, Vaheshta Sethna, Anna Blasi, Eileen Daly, Maria Kuklisova-Murgasova, Sarah Lloyd-Fox, et al. 2019. 'Familial Risk of Autism Alters Subcortical and Cerebellar Brain Anatomy in Infants and Predicts the Emergence of Repetitive Behaviors in Early Childhood'. *Autism Research* 12 (4): 614–27. <https://doi.org/10.1002/aur.2083>.

- [x] **There are many instances when the language use is awkward – surprising given the number of native English speaker co-authors. Some examples (there are others):**

[x] • **Line 51 – mental ill-health**

This is a commonly used term in mental health literature, e.g., Glozier N. Mental ill health and fitness for work. *Occupational and Environmental Medicine* 2002;59:714-720.

[x] • **Line 117 – brain dynamics develop quickly with age – ‘at scan’ is irrelevant.**

We use this to differentiate from PND age and to make it clearer for the reader that might not be familiar with neonatal neuroscience jargon.

[x] • **Line 187 and forward – dwelling time – do they mean dwell time?**

We have changed this accordingly.

[x] • **similarly dwelling transitions? Dwelling sequences??**

We have amended the text.

[x] • Many times, they say ‘later in childhood’, when they are referring to 18months – this is hardly later in childhood, it is barely out of infancy. Otherwise, it implies much longer follow-up.

We have amended the text and specified 18 months of age.

[x] • Line 383 - This chimes with other....

Thank you, we have rephrased to:

“This is concordant with other evidence...”

[x] • Lines 433-434 – meaning is not clear at all to me.

We have amended the text to remove the ambiguous segment: “– which may impact upon the development of higher order networks.”

Reviewer #2

The authors investigated the characteristics of dynamic functional connectivity in a cohort of term-born (n=324) and preterm-born (n=66) neonates. They identified six fundamental transient states of neonatal dynamic functional connectivity and found disrupted brain dynamics in preterm-born neonates. They also found that brain state dynamics at birth appear to be correlated with neurodevelopment outcomes in early childhood. Overall, there is much to applaud in this manuscript. The authors firstly characterize the fundamental properties in early human development. Their findings would guide future work to explore prognostic value and intervention of fundamental dynamic characteristics of early human development.

We are very grateful for the positive appraisal of our manuscript and the constructive feedback.

Some questions and suggestions which may improve the manuscript are provided below:

- [x] **ABSTRACT, line 64, where is the difference between synchronisation and synchrony? Like line168, will using regional constrained states make it clear for readers?**

Thanks for the suggestion, we have edited the abstract as suggested: *“three whole-brain synchronisation states and three regionally **constrained** states...”*

- [x] **Introduction, line 101, what is the motivation for using Kuramoto Order Parameter (KOP) and [[Leading Eigenvector Analysis]] (LEiDA) in this study? What is the advantage of KOP and LEiDA? Why KOP and LEiDA are the state-of art techniques? Although authors suggested the limitations of studies based on sliding window approaches, like hidden (Semi-)Markov models also could be employed this problem.**

Thank you for requesting this clarification. We chose to use KOP and LEiDA methods as both are time-resolved techniques thus making them suitable for exploring brain state dynamics. We have added that information to the text.

*“That is, we identified sub-networks involved in temporal ‘states’, i.e., paroxysmal modes representing synchronisation of the brain, using Leading Eigenvector Analysis (LEiDA) **which is a time-resolved metric...**”*

The reviewer is right to mention other techniques such as Hidden Markov Models (HMMs) could also be employed to study brain dynamics. We have mentioned this in the revised version of our discussion section:

“Thus, the choice of a time-resolved approach like LEiDA is a strength in our methodology, though other techniques such as Hidden Markov Models (HMMs) are also available (Chen et al. 2016).”

- [x] **Results, line 197, “....., with probabilities above 83% for all six states”, however, in Figure 3A, the minimum dwelling transitions probability is 89.75%.**

Apologies for this mistake, it should state “above 89%” as the reviewer suggests. We have amended the text accordingly.

*“Most occurrences are those of dwelling sequences, i.e., repeated continuous occurrences of the same state, with probabilities **above 89%** for all six states.”*

- [] Results, line 198-201, why did 12 most frequent transitions incorporate an indirect transition to and from the high whole-brain synchronization global state A instead of state FP? As for state FP, these transitions also occurred via the intermediate states, B and C.

Thank you for raising this important point. According to our results, global state A is more often accessed through global state B (3.33%) or C (2.89%), but not so frequently through state FP (0.62%). We think that this finding demonstrates an interesting pattern seen in neonatal brain dynamics, which contrast with other states such as FP, which are accessed through a variety of global and regionally constrained states. We have clarified this in lines 198-201:

“Excluding those dwelling sequences, a complex dynamic profile is displayed by the 12 most frequent transitions. For instance, the brain transitions to and from global state A (which has high levels of synchronisation), via global states B and C (which have intermediate levels of synchronisation), while brain transitions into FP state occur through global and regionally constrained states: global state B, occipital state, and SM state.”

- [x] Results, line 231-233, “.....transitions from intermediate whole brain synchronisation state B to the occipital state increased with PMA at scan but decreased with PND at scan.”, however, in Figure 3E, F and Figure 4H, there are no transitions from intermediate whole brain synchronisation state B to the state occipital.

We would like to thank the reviewer for spotting this mistake. We have amended the text:

“Age at scan and postnatal experience had distinct correlates: for example, transitions from intermediate whole-brain synchronisation state C to the occipital state decreased with PMA at scan, while transitions in the opposite direction (from occipital state to synchronisation state C) were increased with PND at scan.”

- [x] Methods, Global dynamics, line 641, how to define Preterm–born for term-born?

Apologies for the confusion, “preterm-born” was set to 0 for term-born individuals and 1 for preterm-born individuals. We have clarified in the revised version of the manuscript.

“Prematurity was coded as a binary variable with 1 for preterm-born individuals (GA at birth less than 37) and 0 for term-born participants (GA at birth of 37 or more)”

Why did authors use PND as dependent variable in GLM1 but not in GLM2?

PND was not included in GLM2 as PND is unavoidably much higher in preterm-born infants in comparison to those born at term, and thus it would not be possible to disentangle the effect from preterm birth.

- [x] In Figure 3C, D, Figure 4E, and Figure 5, authors are need to add the meaning of values in circle to the illustration of figure.

Thank you for noting this. The values shown are t-statistics, which we have now clarified in the respective legends and labels of the revised manuscript.

Reviewer #1 (Remarks to the Author):

The authors did some additional, extensive reanalyses to further support their findings, and moderated text in areas that they did not change.

I have no further comments, apart from a few grammatical corrections in added text:

- high interneuron migrations to these regions
- (though not necessary diagnostic) – should be necessarily.

Reviewer #2 (Remarks to the Author):

The authors have addressed all my concerns. I do not have any further questions

Reviewer #1

The authors did some additional, extensive reanalyses to further support their findings, and moderated text in areas that they did not change.

I have no further comments, apart from a few grammatical corrections in added text:

- high interneuron migrations to these regions**
- (though not necessary diagnostic) – should be necessarily.**

Reviewer #2:

The authors have addressed all my concerns. I do not have any further questions.

We thank both reviewers for the constructive feedback provided, which has improved the final version of our manuscript.

We are grateful for the grammatical corrections provided by reviewer #1, which we have now corrected in the revised version of our manuscript.